# Genetics and environment distinctively shape the human immune cell epigenome

Wenliang Wang[1,18], Manoj Hariharan [1,18], Wubin Ding [1,18], Anna Bartlett [1], Cesar Barragan[1], Rosa Castanon [1], Ruoxuan Wang[1,2], Vince Rothenberg[1], Haili Song[1], Joseph R. Nery [1], Andrew Aldridge[3], Jordan Altshul[1], Mia Kenworthy[1], Hanqing Liu[1], Wei Tian[1], Jingtian Zhou[1], Qiurui Zeng[1], Huaming Chen [1], Bei Wei[4], Irem B. Gündüz [5], Todd Norell[6], Timothy J. Broderick[6], Micah T. McClain[7,8], Lisa L. Satterwhite [9], Thomas W. Burke [7], Elizabeth A. Petzold[7], Xiling Shen[10], Christopher W. Woods[7,8], Vance G. Fowler Jr.[7,11], Felicia Ruffin [7], Parinya Panuwet[12], Dana B. Barr[12], Jennifer L. Beare[13], Anthony K. Smith[13], Rachel R. Spurbeck[13], Sindhu Vangeti[14], Irene Ramos [14], German Nudelman[14], Stuart C. Sealfon [14], Flora Castellino[15], Anna Maria Walley[16], Thomas Evans[16], Fabian Müller [5], William J. Greenleaf [4] & Joseph R. Ecker [1,17] ✉

The epigenome of human immune cells is shaped by both genetics and environmental factors, yet the relative contributions of these influences remain incompletely characterized. Here we use single-nucleus methylation sequencing and assay for transposase-accessible chromatin using sequencing (ATAC–seq) to systematically explore how pathogen and chemical exposures, along with genetic variation, are associated with changes in the immune cell epigenome. Distinct exposure-associated differentially methylated regions (eDMRs) and differentially accessible regions were identified, and a significant correlation between these two modalities was observed. Additionally, genotype-associated DMRs (gDMRs) were detected, indicating that eDMRs are enriched in regulatory regions, whereas gDMRs are preferentially located within gene body marks. Disease-associated single-nucleotide polymorphisms were frequently colocalized with methylation quantitative trait loci, providing cell-type-specific insights into the genetic basis of diseases. These findings highlight the complex interplay between genetic and environmental factors in shaping the immune cell epigenome and advance understanding of immune cell regulation in health and disease.

The debate between nature and nurture is a long-standing discussion in both biology and society. It centers around the relative impact of genetic inheritance (nature) versus environmental factors (nurture) on human development. While inherited epigenetic marks are passed down through generations, acquired features arise from environmental influences and can alter gene expression without changing the underlying DNA sequence. Understanding the contributions of these two sources of epigenetic variation is crucial for comprehending how genes and environments together shape biological outcomes.

The interplay between genetic predispositions and environmental factors shapes biological outcomes[1]. Previous twin studies based on bulk tissues have estimated that the average heritability of methylation levels at cytosine–guanine dinucleotides (CpGs) across the genome ranges from 5% to 19% in different tissues[2–5]. Methylation quantitative trait locus (meQTL) studies have shown associations between genetic

variation and methylation at individual CpG sites[6–11]. However, these studies rely on bulk tissues and often use arrays to profile the methylome. The genome-wide, cell-type-specific relationship between genetic variation and methylation has yet to be fully elucidated.

To investigate the cell-type-specific contributions of genetic and environmental factors on the human immune cell epigenome, we analyzed 171 peripheral blood mononuclear cell (PBMC) samples from 110 individuals with defined exposures to pathogens (human immunodeficiency virus 1 (HIV-1), influenza A virus (IAV), methicillin-resistant *Staphylococcus aureus* (MRSA), methicillin-susceptible *Staphylococcus aureus* (MSSA), anthrax vaccine and severe acute respiratory syndrome coronavirus 2 (SARS-CoV-2)) and chemicals (organophosphates (OPs)). Seven major immune cell types were isolated using fluorescence-activated cell sorting (FACS), followed by snmC-seq2 and single-nucleus ATAC–seq (snATAC–seq) profiling. We identified exposure-associated differentially methylated regions (eDMRs) and genotype-associated DMRs (gDMRs), which showed distinct genomic enrichments—eDMRs were enriched at enhancers, whereas gDMRs were predominantly found in gene bodies, highlighting divergent regulatory mechanisms. Exposure-specific DMRs exhibited unique epigenomic signatures, with several implicating master transcription factor (TF) binding sites. HIV-1-induced changes in DNA methylation and chromatin accessibility were significantly correlated. Moreover, we observed substantial colocalization between meQTLs and GWAS variants, providing cell-type-resolved insights into disease-associated loci. This comprehensive atlas of exposure- and genetics-associated epigenomic features in human immune cells provides a valuable resource for mechanistic studies of infectious and genetic disease.

## Results

### An exposures-driven single-cell epigenomic atlas of human immune cells

We collected 171 PBMC samples from 110 individuals who were exposed or not exposed to seven major exposures (Fig. 1 and Supplementary Table 1). HIV-1-exposed samples were collected from the iPrEx cohort, a phase 3 clinical trial[12], which was designed to evaluate the efficacy of pre-exposure prophylaxis (PrEP) for HIV-1 prevention. We analyzed PBMC samples from nine donors at three distinct time points—approximately 200 days before HIV-1 positivity (pre), the day of HIV-1 diagnosis (acu) and approximately 200 days after initiating treatment (cro). For IAV exposure, the BARDA-Vaccitech FLU010 study[13] evaluated the VTP-100 vaccine against the H3N2 influenza virus strain. We studied prechallenge and 28 days postchallenge (with live H3N2 influenza virus) PBMC samples from 18 donors who received the placebo vaccine. Additionally, we also analyzed PBMC samples from donors who were exposed to SARS-CoV-2 and had severe or nonsevere coronavirus disease 2019 (COVID-19) symptoms.

For bacterial exposure, PBMCs from 19 patients who tested positive for MRSA or MSSA were analyzed, yielding a total of 27 samples. We also obtained PBMC samples from 27 vaccinated participants who handled *Bacillus anthracis* in a controlled Biosafety Level 3 (BSL3) facility while wearing appropriate personal protective equipment. These individuals were trained scientists working in a BSL3 laboratory who had received either BioThrax, an inactivated acellular vaccine primarily composed of the nonpathogenic protective antigen (PA) protein or the anthrax vaccine adsorbed, in accordance with the facility's safety protocols. OPs are a class of pesticides known to have a severe impact on the dopaminergic and serotonergic systems. A common form of this pesticide, that is, chlorpyrifos, has been widely used in the United States. As part of this study, samples were collected from farm workers and nearby residents. By tracking the levels of TCPY over four months, samples from 27 donors were classified as exposed to high, moderate or low levels of OP.

PBMCs from these samples were FAC-sorted into major immune cell types (Extended Data Fig. 1) and processed through the snmC-seq2 pipeline as previously described[14]. A total of 96 cells per sample and cell type were sorted to ensure adequate coverage. To control for potential batch effects, all cell types were sorted onto the same 384-well plate (Extended Data Fig. 1). Single-nucleus methylation data were analyzed using in-house pipelines, as previously described[14]. After filtering out low-quality cells, individual cells exhibited 5–10% genome coverage (Extended Data Fig. 2a). A total of 104,000 cells were then clustered using the average CG methylation score in 5-kb bins across the autosomes. Pseudobulk profiles demonstrated high genome coverage and sufficient depth at single-CpG resolution (Extended Data Fig. 2b–e). snATAC–seq data from our companion study were integrated with the methylation data and cell-type labels were transferred.

### Heterogeneity in methylation profiles reveals exposure-specific immune cell clusters

To assess whether exposures are associated with the methylome of each immune cell type, we performed within-cell-type clustering for the major sorted immune cell types. The t-distributed stochastic neighbor embedding (t-SNE), with or without harmony integration, showed similar embedding (Extended Data Fig. 3a). This analysis revealed heterogeneity in methylation profiles, resulting in more than ten distinct clusters for each cell type (Fig. 2a). We observed significant bias of cells from each exposure in each cell type. Of note, HIV-1, SARS-CoV-2 and MRSA/MSSA exposures were associated with distinct monocyte, CD4 and CD8 naive T cell profiles (Fig. 2a and Extended Data Fig. 3b). The proportion of immune cells from each exposure in these clusters varied substantially (Fig. 2b). For example, we observed a cluster of monocytes enriched in both severe and nonsevere COVID samples. Considering that the biased distribution among clusters of different exposures might also be caused by heterogeneity between individuals rather than the specific exposures, we focused on proportional changes in HIV-1 and IAV samples collected from the same donors before and after infections. While IAV samples have comparable proportions in each cluster across the cell types, HIV-1 infection markedly changed the cell proportions among clusters, indicating that HIV-1 remodeled the global methylome and functional states of these immune cells, particularly NK cells, CD8 memory and naive T cells (Fig. 2c).

To further validate the changes in functional states of different immune cell types associated with HIV-1 infection, we performed within-cell-type clustering of HIV-1 samples. The clustering revealed that cells from the three stages of HIV-1 infection (pre, acu and cro) were unevenly distributed among the clusters, with biased clusters exhibiting different global methylation levels (Fig. 2d and Extended Data Fig. 4a). To further characterize the identity of each subtype that is significantly differentially distributed among the three stages (false discovery rate (FDR) < 0.05, Fisher's exact test), we identified differentially methylated genes (DMGs) in each cluster compared to all other clusters and performed functional enrichment analysis of these genes. Immune cell activation and differentiation-related functions are enriched among these clusters (Fig. 2e). The CD8 memory T cell cluster, specifically enriched in 'acu' stage cells, is enriched in positive regulation of immune response.

The monocyte cluster uniquely enriched with COVID samples and depleted in controls and other exposures is likely associated with this specific exposure (Fig. 2f). Almost half of the monocytes from both severe and nonsevere COVID-19 samples are separated in these two clusters (Fig. 2f), significantly more than the control samples ($P = 2.05 \times 10^{-237}$, Fisher's exact test). Gene body methylation levels are highly similar between classical and nonclassical monocyte markers (Fig. 2g). We identified 321 DMGs between 'monocyte1' and 'monocyte2' (Extended Data Fig. 4b). Functional enrichment of these genes showed that genes hypomethylated in both monocyte clusters are enriched in pro-inflammation functions like 'IL-18 signaling pathway' and 'regulation of interleukin-1 (IL-1) production' (Fig. 2h). Both IL-18 and IL-1 have been reported to have protective roles during mouse coronavirus infection[15], whereas IL-1 has a pivotal role in the induction of cytokine

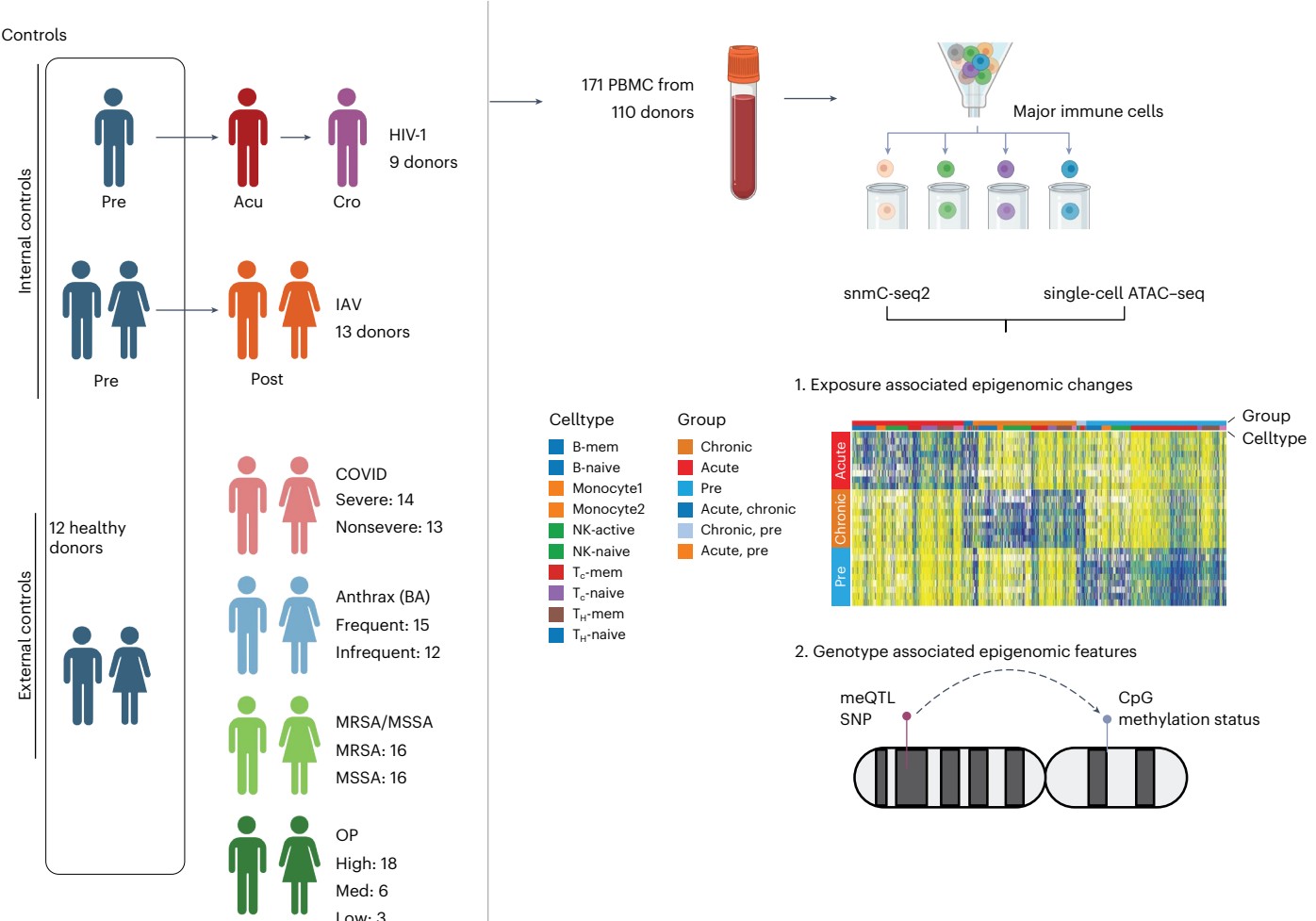

**Fig. 1 | Overview of the study.** For HIV-1 and IAV, we have internal control samples that are from the same set of donors before infection and collected samples from them after exposure. We also collected PBMC from 12 healthy donors as external controls. For exposures without internal controls (COVID, anthrax vaccine, MRSA/MSSA and OP), all healthy samples were used as controls. We performed snATAC–seq and snmC-seq2 on the PBMCs and identified the eDMRs associated with exposures and genotypes. We also identified the gDMRs using this dataset. BA, *B. anthracis.* The figure is created with BioRender.com.

storm as a result of uncontrolled immune responses in SARS-CoV-2 infection[16]. Moreover, in addition to IL-1 and IL-18 production-related functions, DMGs in 'monocyte2' are also enriched for 'phagocytosis' and 'endocytosis' (Fig. 2h), indicating the antigen presentation function of this cluster, which is specifically enriched in COVID-19 monocytes.

### Cell-type-specific epigenomic responses to exposures

We refined our cell types by integrating methylation information with FACS. B cells and NK cells can be separated into two clusters by global CG methylation level (Fig. 3a). The higher methylated clusters were annotated as a naive state (B-naive and NK-naive), while the cluster with lower methylation level of B cells is annotated as memory B cells (B-mem) and a lower methylated cluster of NK cells is defined as the active state (NK-active)[17]. We observed some inconsistency between FACS and methylation profile clustering, with some naive T cells clustered together with memory T cells, indicating a transition of cell states in some naive T cells (Fig. 3a). These cells were still labeled as naive T cells based on cell-surface marker CCR7+ and CD45RA+. Some sorted NK cells (CD56+ or CD16+ and CD14−) exhibited methylation profiles similar to monocytes, suggesting that they may represent CD16+ monocytes. These cells were excluded from the downstream analysis. We also observed bias among different exposures in the Uniform Manifold Approximation and Projection (UMAP) representation (Extended Data Fig. 5a), which may reflect exposure-specific effects. Nevertheless, our clustering results

remained highly consistent following integration-based correction for donor and batch effects (Extended Data Fig. 5b,c). In the following analysis, we dissected the contributions of exposures and genetics to the epigenome in these nine immune cell types.

We aimed to dissect the impact of different exposures on the epigenome in these cell types by identifying the eDMRs. Given the difficulty of controlling other exposures each individual may have experienced and the donors' diverse genomes, we used internal controls for HIV-1 and IAV exposures, for which we have cells before and after exposure. We used a stringent pipeline to identify the eDMRs of other exposures (Methods), in which the external controls are used. These eDMRs were not significantly different between any two sets of controls, minimizing the contribution of individual genetic variation and baseline exposures to these eDMRs.

We identified 756,575 eDMRs across all exposures and cell types, including 517,698 hypomethylated and 238,877 hypermethylated eDMRs (Fig. 3c and Supplementary Table 2). On average, each exposure and cell type exhibited approximately 10,000 eDMRs. SARS-CoV-2, OP and MRSA/MSSA showed the most abundant eDMRs across the majority of the cell types, highlighting their pronounced impact on the epigenetic profiles (Fig. 3c). To control for the false positive rate, we performed sample label shuffling and found that our identified eDMRs were substantially more significant than DMRs detected in shuffled samples (Extended Data Fig. 5d). Of note, the majority of eDMRs in each exposure and cell type consist of single CpG sites (Extended Data Fig. 5e).

We examined the effect-size distributions of exposure-associated eDMRs to assess the magnitude of methylation change (Extended Data Fig. 5f–k). HIV-1 induced broad, multimodal shifts, particularly during early infection. MRSA/MSSA and anthrax vaccine exposures had distinct peaks, while OP caused highly variable changes. COVID responses were skewed by severity and IAV showed a symmetric, trimodal pattern. These distributions illustrate the magnitude of methylation changes associated with each exposure.

Although the externally controlled exposures involved sizable cohorts, donor variability in genetic background, sex and age could still confound the results. To address this, methylation levels were adjusted for these covariates and eDMRs significantly associated with them were excluded, resulting in 271,592 hypomethylated and 138,551 hypermethylated eDMRs that more likely reflect exposure-specific effects. To examine genetic-ancestry-related responses, we identified genetic-ancestry-related eDMRs after regressing out sex and age. Notably, in both MRSA/MSSA and SARS-CoV-2 exposures, individuals of African genetic ancestry showed stronger methylation effects (Fig. 3d,e), with most eDMRs unique to each group (Fig. 3f,g), consistent with reports of greater disease severity in individuals of African genetic ancestry Americans[18–20].

We further examined the association between eDMRs and histone modification peaks from ENCODE[21] in each cell type. We found that both hypomethylated and hypermethylated eDMRs across all exposures are slightly enriched in heterochromatin regions marked by H3K9me3 (Fig. 3h and Extended Data Fig. 6a). Notably, COVID-19-associated hypermethylated eDMRs in monocytes, naive T cells and NK-naive cells showed significant enrichment in enhancer marks like H3K27ac and H3K4me1, as well as the promoter mark H3K4me3, across all immune cell types (Fig. 3h). Similar patterns were observed for HIV-1, with hypomethylated eDMRs in CD8 memory T cells and hypermethylated eDMRs in CD8 naive T cells showing enrichment in these active histone marks (Fig. 3h). Furthermore, MSSA monocyte hypermethylated eDMRs were enriched in H3K27ac. In contrast, NK-active cells from MSSA-infected individuals (but not MRSA-infected ones) showed enrichment in enhancer regions. Interestingly, naive T cells from severe COVID-19 patients had hypomethylated eDMRs with greater enrichment in active histone marks than those from nonsevere patients (Fig. 3h). These eDMRs are generally depleted at CG islands, promoters and short interspersed nuclear elements but slightly enriched in DNA, long interspersed nuclear elements and long terminal repeat transposable elements (Extended Data Fig. 6b).

DNA methylation influences TF binding to DNA[22,23], potentially altering gene expression. We observed that master TF motifs were enriched in COVID-19 monocytes and CD4 naive T cells. For example, PRDM1 (encoding BLIMP-1) in monocytes[24] and TCF family TFs (including TCF7) motifs in CD4 naive T cells are uniquely enriched in COVID-19 samples[24,25]. Similarly, CEBP family TF motifs are specifically enriched in MRSA/MSSA monocytes[26] (Fig. 3i). These results

suggest that methylomes associated with exposures might be able to inhibit the binding of master TFs. On the other hand, hypomethylated eDMRs across the exposures share many similarly enriched motifs (Extended Data Fig. 6c). Although further validation is needed, ChIP-seq profiles from previous studies[27] showed significant enrichment of eDMRs within relevant TF binding sites (Extended Data Fig. 6d). To assess functional relevance, we examined their overlap with differentially expressed genes from prior HIV-1 and SARS-CoV-2 studies[28,29] and found significant enrichment, especially in monocytes and T cells, compared to random genes or CpGs (Extended Data Fig. 6e–h).

## Linking DNA methylation and chromatin accessibility in exposures

We integrated snATAC–seq data from paired and unpaired donors across all exposure groups, except for MRSA and MSSA. For the HIV-1 exposure, in which samples were collected at different time points from the same donors, we integrated snDNA methylation data with snATAC–seq data using 5-kb bins on autosomes. UMAP plots generated before integration showed clear batch effects between the two modalities, which were partially mitigated following integration (Extended Data Fig. 7a,b). We mapped cells from single-cell ATAC–seq to methylation clusters and transferred cell-type labels via canonical correlation analysis[30] (Fig. 4a,b). We observed a strong genome-wide correlation between the two modalities across all cell types, with the highest correlation observed in monocytes (Extended Data Fig. 7c). Interestingly, we detected a loss of methylation and an increase in accessibility after HIV-1 infection in memory CD8 T cells at the intron of DGKH (Fig. 4c), a gene previously reported to exhibit differential methylation between elite controllers and individuals receiving antiretroviral therapy[31]. Although this region experienced a loss of chromatin accessibility, the methylation level remained unchanged between the 'acute' and 'chronic' stages (Fig. 4c). We found a considerable fraction of eDMRs are consistent with changes in chromatin accessibility (Fig. 4d). The highest overlap (25.6%) between these two modalities was observed in 'pre' stage hypo-eDMRs in CD8 naive T cells.

We extended our analysis to additional exposures and found that exposure-associated differentially accessible regions overlapped with corresponding exposure-associated eDMRs by approximately 1–6% (Extended Data Fig. 7d–g). The lower concordance between the two modalities—compared to what we observed for HIV-1—may be attributed to the use of internal controls in both assays for the HIV-1 samples. These findings underscore the importance of longitudinal data for robust identification of exposure-associated epigenomic changes.

## Genotype-associated DMRs reveal the influence of genetics on immune cell epigenomes

We also used this single-cell epigenomic atlas to identify gDMRs. We identified the SNPs of each individual using the DNA methylation data with biscuit[32], followed by imputation and filtering (Methods). To

**Fig. 2 | Within-cell-type changes are associated with each exposure. a**, UMAP of cells in each cell type from FACS using snmC-seq2 data. The cells are colored according to exposure (HIV, exposure to HIV; HIV_pre, before HIV infection from the same donors; Flu_pos, after IAV infection; Flu_pre, before IAV infection from the same donors; COVID_S, severe COVID patient samples; COVID_nS, nonsevere COVID patient samples; MRSA, samples exposed to MRSA; MSSA, samples exposed to MSSA; BA, samples from the donors that have taken anthrax vaccine and work frequently or infrequently in a controlled BSL3 facility handling *B. anthracis*; OP, samples from the donors that are exposed to OP). **b**, Bar plots show the proportions of cells from each group and cell type in the Leiden clusters, colored by the FACS cell types. The *x* axis is the Leiden clusters and the *y* axis shows the proportions of cells in each Leiden cluster in each group. **c**, Scatter plot shows the cell proportional changes before and after infection with HIV and IAV in each Leiden cluster. Dots represent clusters with IAV exposures and crosses indicate HIV exposures. Color shows the FACS cell types. **d**, UMAP of cells

from HIV exposure donors in the cell types that have the most cell proportion changes. The three rows are UMAP of cells from NK cell, $T_c$-mem and $T_c$-naive, and the columns are the cells from 'pre', 'acu' and 'cro' stages. Cells from the stage are shown in red and cells from other stages are shown in gray. **e**, The dot plot shows the odds ratio using two-sided Fisher's exact test on cells from the three HIV infection stages ('pre', 'acu', 'cro') in the Leiden clusters. The heatmap shows the GO enrichment of DMGs of the corresponding Leiden cluster. The dot plot and heatmap have the same *x* axis. **f**, Two clusters of monocytes were identified. The bar plot depicts the cell proportions of the two monocyte clusters in controls, severe and nonsevere COVID-19 samples. Statistical tests were done using the two-sided chi-square test. (Asterisk indicates that $P = 1.32 \times 10^{-278}$). **g**, Gene body methylation levels at classical and nonclassical monocyte markers in the two clusters of monocytes. **h**, GO enrichment using the hypergeometric test with Benjamini–Hochberg FDR correction (implemented in Metascape) for DMGs between the two clusters of monocytes.

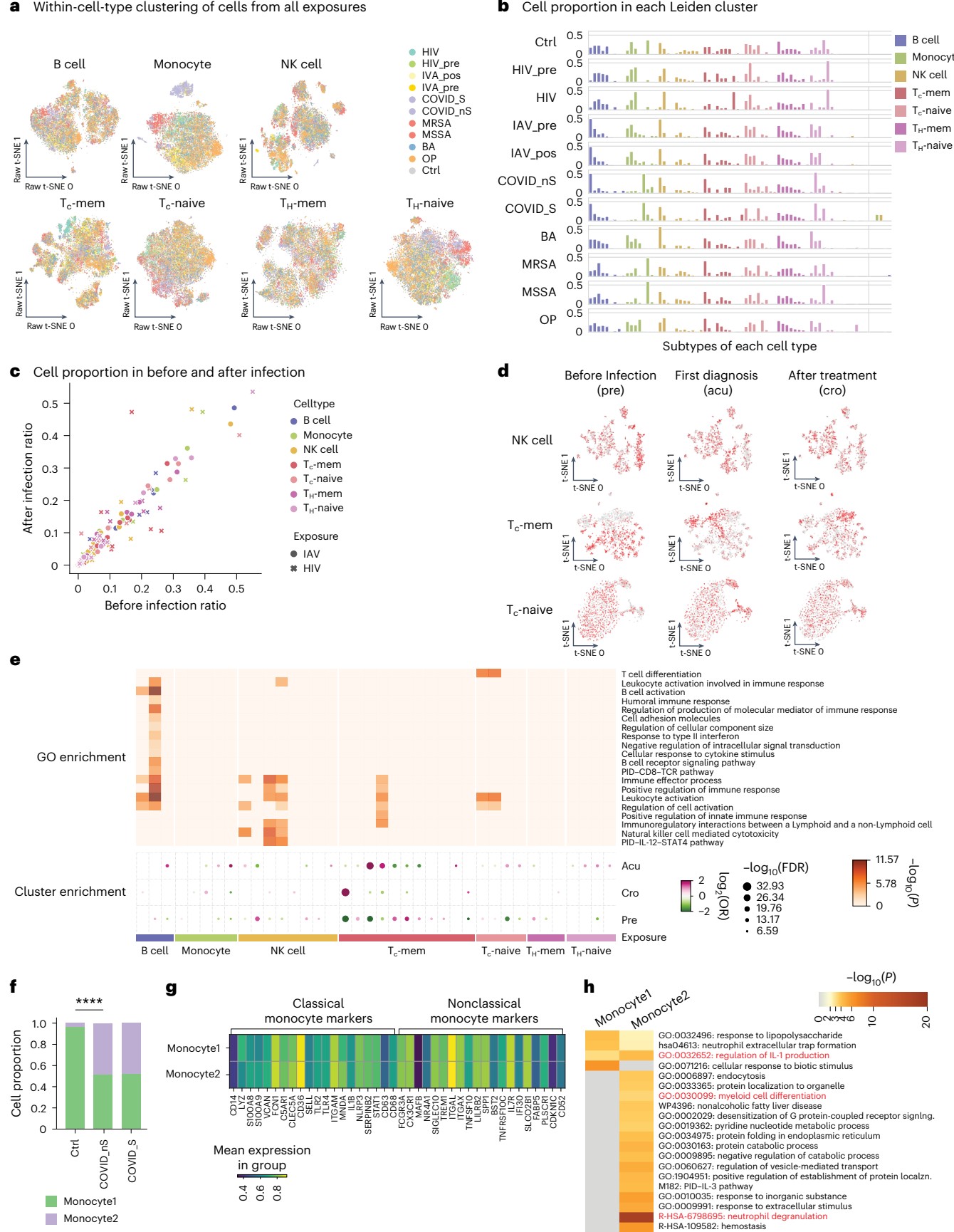

confirm the accuracy of the SNP calls obtained using this strategy, we compared the SNPs from whole-genome sequencing (WGS) and methylation data from a previous study[33]. The results showed a strong agreement with WGS SNPs in a 10-Mb region, with low rates of false positives and false negatives (Extended Data Fig. 8a) and high correlation in alternate allele frequency at the overlapped SNPs (Extended Data Fig. 8b). Running SNP calls on methylation on GM12878 also show high accuracy compared to the true set (Extended Data Fig. 8c).

We first identified differentially methylated regions (DMRs) at CpG sites within each cell type and quantified methylation levels for each individual. DMRs and SNPs were filtered, followed by performing meQTL analysis as described (Methods). Principal component analysis (PCA) of the genotypes further confirmed the high quality of the SNPs derived from bisulfite reads (Extended Data Fig. 8d).

After stringent filtering of the meQTL–DMR pairs (Supplementary Tables 3–11), we identified 275,283 gDMRs across all nine cell types, of which 214,933 were *cis*-correlated with SNP and 60,350 were *trans*-correlated (Supplementary Table 12). The number of *cis*-gDMRs is comparable across different cell types except CD8 memory T cells, which have more gDMRs compared to other cell types (Fig. 5a). The *trans*-gDMRs are mostly identified in various T cell types (Fig. 5a). The *cis*-gDMRs and *trans*-gDMRs in T cells are mostly single-CpG sites (Extended Data Fig. 8e). The number of gDMRs is correlated with the number of cells sequenced in each cell type (Extended Data Fig. 8f). We did an enrichment analysis between these two sets of DMRs on different histone marks. In contrast to eDMRs at enhancer marks, gDMRs are predominantly enriched at gene body mark H3K36me3 peaks, especially in memory state lymphocytes (B cell, CD4 and CD8 T cells; Fig. 5b), while eDMRs are enriched at enhancer and promoter regions in naive lymphocytes. We also did enrichment analysis of them at the loop anchors in different immune cells from ENCODE, which showed that gDMRs are more enriched at loop anchors in memory state lymphocytes and eDMRs are more associated with naive lymphocytes (Fig. 5c). This indicates that eDMRs and gDMRs might regulate gene expression in different cell types through distinct mechanisms. This differential enrichment underscores the complexity of epigenetic control, with eDMRs and gDMRs contributing uniquely to the gene-expression landscape depending on the cell type and the nature of the environmental or genetic input. Both eDMRs and gDMRs exhibit similar genomic feature enrichment regarding genes and transposable elements (Extended Data Fig. 8g).

We performed a functional enrichment analysis on genes with DMRs that overlap H3K36me3 peaks. Both eDMRs and gDMRs were significantly enriched in housekeeping and immune-related functions, with immune functions more significant in eDMRs (Fig. 5d). This distinction highlights the specific impact of environmental and genetic factors on immune-related gene regulation through epigenetic modifications.

Besides chromatin states, the enriched motifs also differ between eDMRs and gDMRs. While both hypomethylated and hypermethylated eDMRs are enriched mainly in immune-related TF motifs, gDMRs do not exhibit enrichment of immune TF motifs (Extended Data Fig. 8h). Using gDMRs as the background in Homer analysis, we identified significant enrichment of RUNX and ETS family TF motifs—including PU.1, ETS1 and Fli1—within eDMRs. In contrast, no significant motif enrichment, except for ETS motifs, was observed in the gDMRs using eDMRs as background in CD8 memory T cells. These results suggest that eDMRs are primarily enriched with key TF binding sites in immune cells, potentially having a role in regulating gene expression through TF binding.

## Cell-type-specific colocalization of gDMRs and GWAS SNPs links to immune diseases

To investigate the association of gDMRs with human diseases and immune-related traits, we performed colocalization analysis between our meQTLs and GWAS SNPs (Methods) linked to various traits (Supplementary Table 13). We identified many colocalized GWAS SNPs and meQTLs within these immune cell types, suggesting potential cell-type-specific regulatory connections between methylation changes and genetic variants associated with immune functions. Enrichment analysis of GWAS SNPs within the meQTLs of each cell type revealed a predominant enrichment in CD8 naive T cells, as well as in CD4 memory and naive T cells (Extended Data Fig. 9a). This suggests that these specific immune cell types are particularly influenced by genetic variants associated with immune-related traits and diseases. For example, GWAS SNPs associated with Gallstone disease and Eczema are enriched in CD8 naive T cells and CD4 memory T cells meQTLs that colocalize with the GWAS SNPs (Extended Data Fig. 9a). We further linked the gDMRs to specific diseases and phenotypes (Extended Data Fig. 9b) through colocalization analysis between meQTLs and phenotype-associated GWAS SNPs. The analysis revealed that most gDMRs are associated with only a single phenotype (Extended Data Fig. 9b). Gallstone disease and Eczema showed the highest number of associated gDMRs, particularly in T cells, highlighting the potential role of T cell–specific epigenetic regulation in mediating genetic risk for these immune-related diseases. We further linked the gDMRs with gene expression (Methods) by performing summary-based Mendelian randomization (SMR)[34] analysis and identified 252,598 substantial associations (Supplementary Table 14).

This analysis enabled us to uncover cell-type-specific regulatory connections between gDMRs and various diseases and phenotypes. For instance, SNPs associated with eczema disease showed colocalization with multiple meQTLs across various immune cell types in a cell-type-specific manner. Most of the GWAS SNPs only colocalize with meQTL in one cell type (Fig. 6a), indicating a high degree of cell-type specificity in the regulatory effects of these GWAS SNPs on the epigenome. This suggests that the impact of genetic variants on DNA methylation, and consequently on gene regulation, can be highly specialized and confined to particular immune cell types. Notably, the top eczema-associated SNP, rs10791824, colocalizes with both a meQTL and an expression QTL (eQTL; Fig. 6b). SMR analysis further reveals that the SNP-associated DMR is significantly correlated with the expression of the *EFEMP2* gene, which has been implicated in eczema pathogenesis. This cell-type-specific colocalization information will greatly facilitate the mechanistic studies on the diseases.

## Discussion

Our study provides a comprehensive, exposure-driven atlas of human immune cells, revealing how genetic and environmental factors shape

---

**Fig. 3 | Identification of eDMRs and their features. a**, UMAP of all cells from all exposures using single-nucleus methylation profiles. Cells are colored by the global methylation level of each cell. **b**, UMAP of all cells from all exposures using single-nucleus methylation profiles. Cells are colored by the FACS cell types. **c**, The bar plots show the hypomethylated (top plot) and hypermethylated (bottom plot) eDMR counts. 'Hypo' indicates the eDMRs are hypomethylated in exposures compared to controls. 'Hyper' is the other way around. The colors represent the exposures and the *x* axis denotes cell type, which is colored and sorted in the same order for all exposures. **d**, Scatter plot shows the two top PCs of ethnicity-dependent MRSA/MSSA eDMRs. **e**, Venn diagram shows the overlap of ethnicity-dependent MRSA/MSSA eDMRs between individuals of African and European genetic ancestry. **f**, Scatter plot shows the two top PCs of ethnicity-dependent COVID-19-associated eDMRs. **g**, Venn diagram shows the overlap of ethnicity-dependent COVID-19-associated eDMRs between individuals of African and European genetic ancestry. **h**, Dot plot shows the enrichment of eDMRs from COVID, HIV and MSSA in histone modification peaks. Each column shows the hypo-eDMRs in that condition. The color of the dots shows the enrichment or depletion in the corresponding histone modification. **i**, Dot plot shows the motif enrichment from HOMER, which uses the one-sided cumulative hypergeometric test on eDMRs from each exposure and cell type. The dot size indicates the *P* values of enrichment and the color shows the cell type from which the eDMRs are.

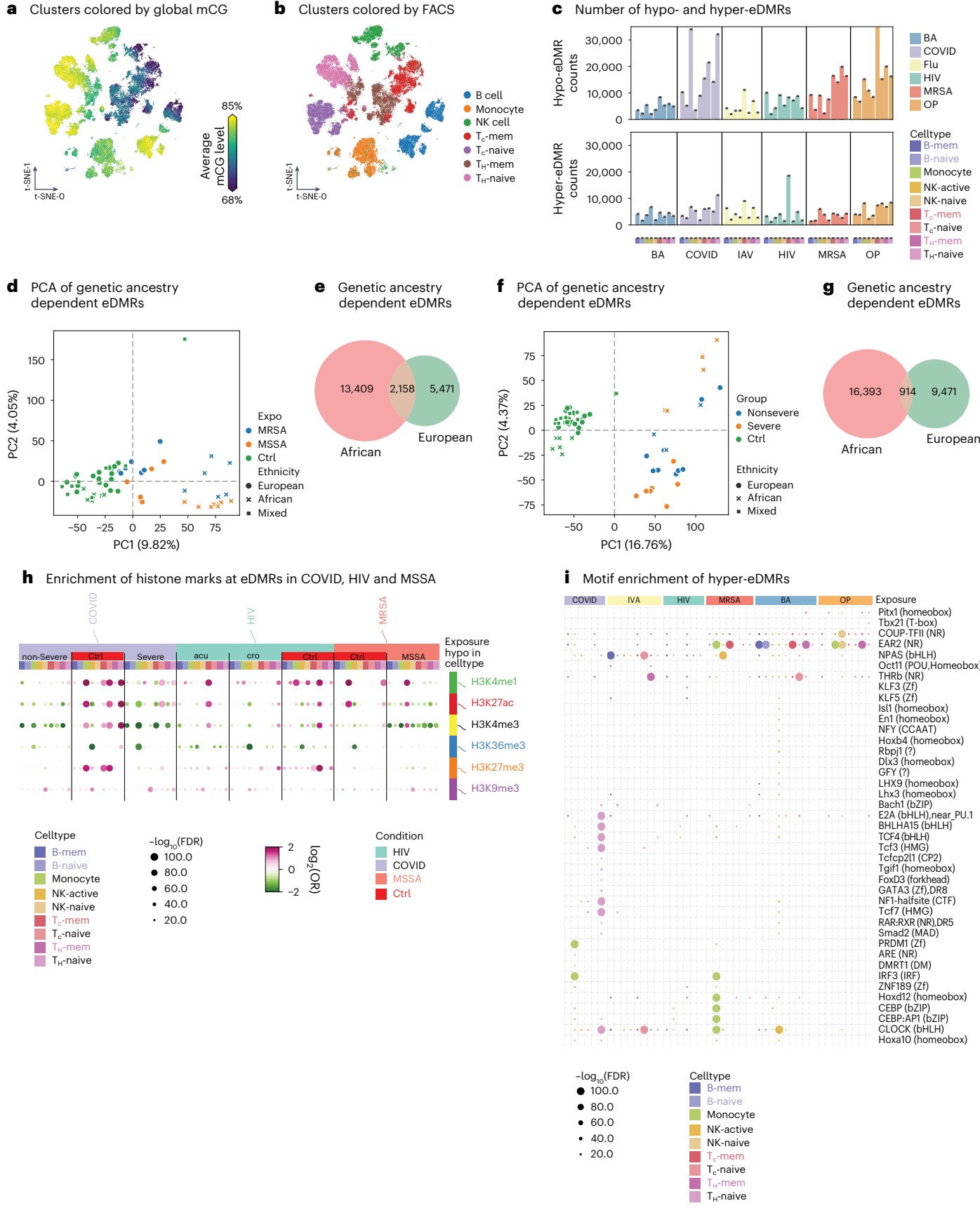

a Clusters colored by global mCG
b Clusters colored by FACS
c Number of hypo- and hyper-eDMRs
d PCA of genetic ancestry dependent eDMRs
e Genetic ancestry dependent eDMRs
f PCA of genetic ancestry dependent eDMRs
g Genetic ancestry dependent eDMRs
h Enrichment of histone marks at eDMRs in COVID, HIV and MSSA
i Motif enrichment of hyper-eDMRs

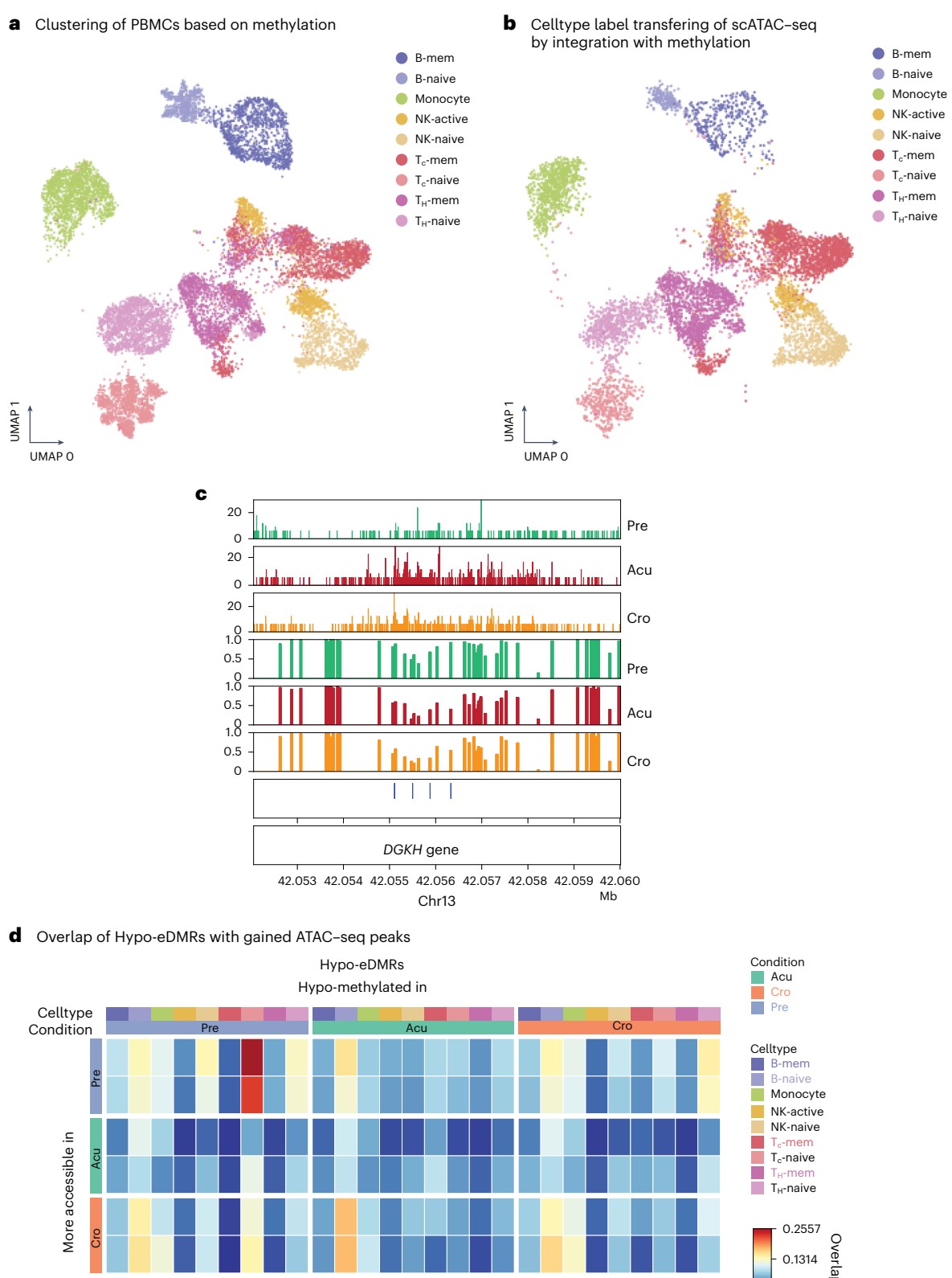

**Fig. 4 | Consistent changes in DNA methylation and chromatin accessibility in HIV exposure. a**, UMAP of cells from HIV exposure using single-nucleus methylation when integrating with snATAC–seq data. The color shows the cell types identified from FACS and DNA methylation. **b**, UMAP of cells from one HIV donor sample after integration with single-nucleus methylation data. The color indicates the cell-type labels transferred from DNA methylation data in integrating the two modalities. **c**, A genome browser view of a region at the *DGKH* gene that has consistent changes after HIV exposure in DNA methylation and chromatin accessibility in CD8 memory T cells. The top three panels are normalized ATAC–seq reads and the DNA methylation panels show the methylation levels in each bin at this locus. The eDMRs are shown in the blue bars. **d**, The overlap between hypo-eDMRs and gained ATAC–seq peaks in each condition in the corresponding cell types. The more accessible peaks are from pairwise comparisons, so each condition has two comparisons. The color of the heatmaps shows the proportion of overlaps between hypo-eDMRs and gained peaks.

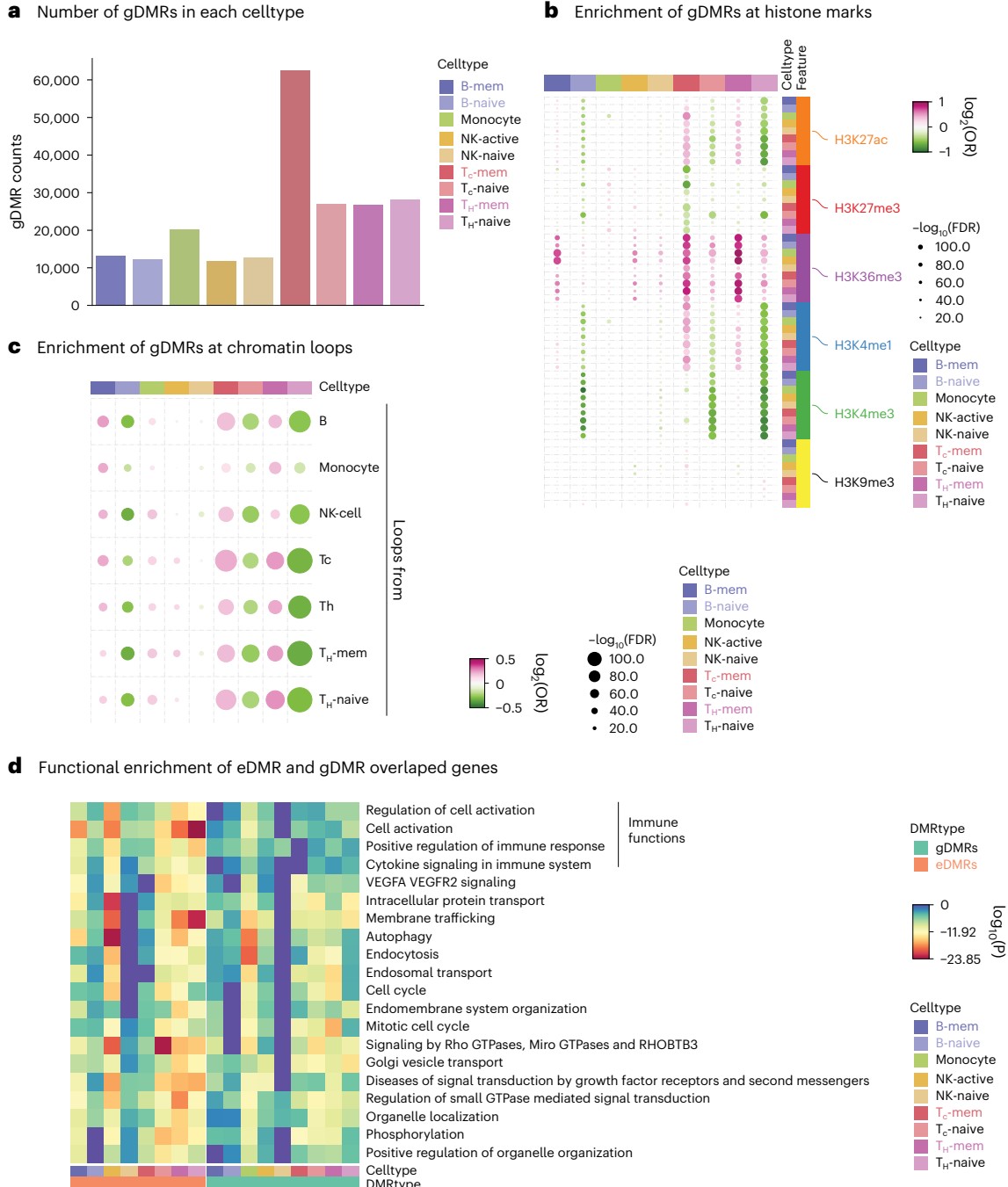

**Fig. 5 | Identification of gDMRs and their features. a**, Bar plot shows the counts of gDMRs in each cell type. The colors of the bars show the cell types. **b**, Dot plot shows the enrichment from Fisher's exact test of gDMRs in histone modification peaks from each cell type. Each column represents a cell type and each row contains one histone modification peak for that cell type. The color of the dots indicates the enrichment or depletion of gDMRs in the corresponding histone modification. **c**, Dot plot shows the enrichment using Fisher's exact test of gDMRs in chromatin loops. Each column is a cell type and each row is the chromatin loop in each cell type. The color of the dots shows the enrichment or depletion of gDMRs in the corresponding chromatin loop. **d**, Metascape using one-sided hypergeometric test, GO enrichment of the genes that overlap with eDMRs or gDMRs. The color of the heatmap shows the $\log_{10}(P)$ of the enrichment. Results for eDMRs and gDMRs in all cell types are sorted in the same order.

the epigenomes. We constructed an intricate epigenomic atlas using snmC-seq2 and ATAC-seq, revealing the epigenomic features associated with different exposures. This comprehensive approach enabled us to identify eDMRs and gDMRs, dissecting the roles of these two factors in shaping the epigenome of human immune cells. We also identified significant colocalization between meQTLs and GWAS-associated disease SNPs, uncovering potential cell-type-specific epigenetic mechanisms underlying these SNPs.

Genetic factors and exposome have long been recognized to shape epigenomes[1]. While previous studies have identified DNA methylation changes associated with genetic[1,6–8] or environmental exposures[1,35,36], our study uniquely examines both eDMRs and gDMRs within the same group of donors. This approach allows for a more reliable comparison of changes driven by these two factors. The differential enrichment of eDMRs and gDMRs on the chromatin indicates that genetics and environments may regulate gene expression differently. Although we

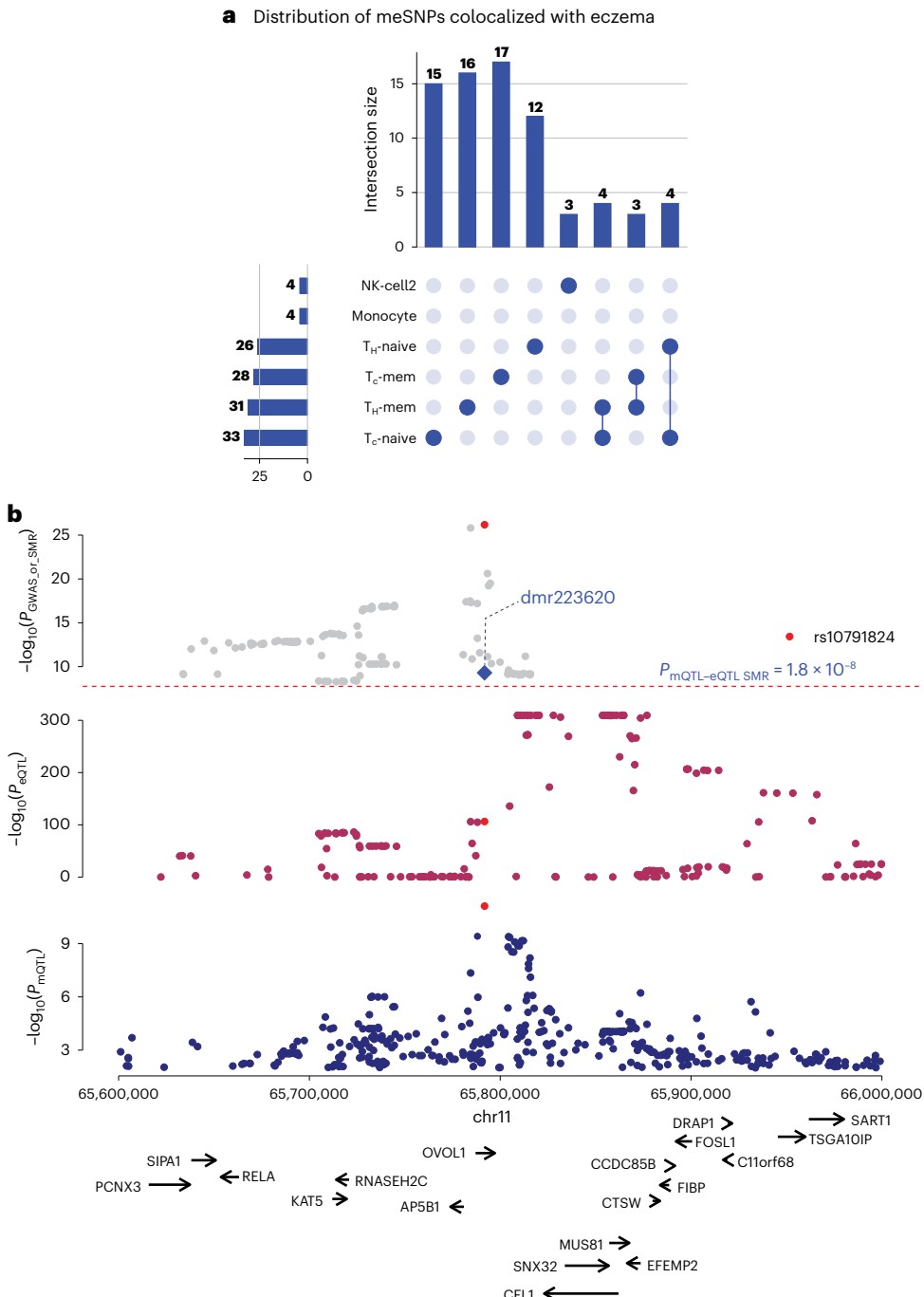

**a** Distribution of meSNPs colocalized with eczema

**Fig. 6 | Colocalization of gDMRs with immune-related disease (eczema) GWAS SNPs. a**, Upset plot shows the number of meQTLs that are colocalized with GWAS SNPs in each cell type or multiple cell types. **b**, Genotype–disease association *P* values in the eczema GWASs (top) at EFEMP2 locus, eQTL (middle) associated with EFEMP2 expression and meQTL signal associated with a DMR in naive T cell (bottom), *P* values are from GWAS associations (Wald test), eQTL test (empirical *P* values via permutations) and meQTL test (empirical *P* values via permutations).

cannot fully disentangle the effects of genetics from different exposure histories in our donors, genetic factors exert a stronger regulatory influence on the gene body in memory lymphocytes. In contrast, exposome regulation has a more pronounced effect on enhancers/promoters in naive lymphocytes.

Our study highlights the intricate contributions of both genetic and environmental factors in shaping the immune cell epigenome. We identified distinct, exposure-specific eDMRs that reflect how immune cells respond to various pathogens and chemicals, while accounting for genetic background. These findings improve our understanding of immune regulation and provide a valuable resource for investigating

the molecular mechanisms of environmental exposures. The eDMRs may also serve as biomarkers for specific exposures, with potential diagnostic and therapeutic applications. However, the mechanisms by which genetics and environment interact to influence the epigenome remain incompletely understood. Addressing this gap will be essential for fully elucidating how gene–environment interactions shape immune function and contribute to disease, fully addressing the 'nature and nurture' question in human disease.

Our study has several limitations. Some exposure groups had relatively small sample sizes, which may limit statistical power and generalizability. In addition, the absence of information on other

environmental exposures or longitudinal data for certain exposures restricts our ability to assess temporal dynamics and causality. Future studies with larger, more balanced cohorts and longitudinal designs will be essential to validate and extend our findings.

## Online content

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

[1]Genomic Analysis Laboratory, The Salk Institute for Biological Studies, La Jolla, CA, USA. [2]Bioinformatics and Systems Biology Program, University of California, San Diego, La Jolla, CA, USA. [3]Duke University School of Medicine, Bryan Research Building, Durham, NC, USA. [4]Department of Genetics, Stanford University, Stanford, CA, USA. [5]Integrative Cellular Biology & Bioinformatics Lab, Saarland University, Saarbrücken, Germany. [6]Healthspan, Resilience, and Performance, Florida Institute for Human and Machine Cognition, Pensacola, FL, USA. [7]Center for Infectious Disease Diagnostics and Innovation, Division of Infectious Diseases, Duke University Medical Center, Durham, NC, USA. [8]Durham Veterans Affairs Medical Center, Durham, NC, USA. [9]Department of Civil and Environmental Engineering, Pratt School of Engineering, Duke University, Durham, NC, USA. [10]Terasaki Institute for Biomedical Innovation, Los Angeles, CA, USA. [11]Duke Clinical Research Institute, Durham, NC, USA. [12]Gangarosa Department of Environmental Health, Rollins School of Public Health, Emory University, Atlanta, GA, USA. [13]Battelle Memorial Institute, Columbus, OH, USA. [14]Department of Neurology, Icahn School of Medicine at Mount Sinai, New York City, NY, USA. [15]U.S. Department of Health and Human Services, Administration for Strategic Preparedness and Response, Biomedical Advanced Research and Development Authority, Washington, DC, USA. [16]Barinthus Biotherapeutics, Germantown, MD, USA. [17]Howard Hughes Medical Institute, The Salk Institute for Biological Studies, La Jolla, CA, USA. [18]These authors contributed equally: Wenliang Wang, Manoj Hariharan, Wubin Ding. ✉e-mail: ecker@salk.edu

## Methods

### Data generation

**FACS of immune cell types.** Cells were sorted into 384-well plates using FACS based on their specific antibody labeling. The FACS antibody cocktail allowed for the identification of seven different immune cell types in blood (Extended Data Fig. 1). The sorted cell types included naive helper T cells (CD3$^+$, CD4$^+$, CCR7$^+$, CD45RA$^+$), memory helper T cells (CD3$^+$, CD4$^+$, CD45RA$^-$), naive cytotoxic T cells (CD3$^+$, CD8$^+$, CCR7$^+$, CD45RA$^+$), memory cytotoxic T cells (CD3$^+$, CD8$^+$, CD45RA$^-$), B cells (CD3$^-$, CD19$^+$), monocytes (CD3$^-$, CD19$^-$, CD14$^+$), NK cells (CD3$^-$, CD19$^-$, CD14$^-$, CD16$^+$, CD56$^+$) and other cells (CD3$^-$, CD19$^-$, CD14$^-$, CD16$^-$, CD56$^-$). The SONY Multi-Application Cell Sorter LE-MA900 Series was used to isolate single cells in 384-well PCR plates containing protein kinase. After cell sorting, the plates were centrifuged to collect the cells at the bottom of the wells, and the wells were then subjected to thermocycling at 50 °C for 20 min. The plates containing the DNA from the cells were subsequently stored at −20 °C or moved directly to library preparation.

**snmC-seq2 library preparation and Illumina sequencing.** For library preparation, we followed the previously described methods for bisulfite conversion and library preparation in snmC-seq2 (refs. [14],[37]). The snmC-seq2 libraries generated from isolated immune cells were sequenced on an Illumina Novaseq 6000 using S4 flow cells in 150-bp paired-end mode. Freedom EVOware (v2.7) was used for library preparation, while Illumina MiSeq control software (v3.1.0.13) and NovaSeq 6000 control software (v1.6.0) and Real-Time Analysis (RTA; v3.4.4) were used for sequencing.

**snATAC–seq.** snATAC–seq was performed as previously described[38], using either the Chromium Next GEM Single Cell ATAC Library & Gel Bead Kit v1.1 (10x Genomics, 1000175) with the Chromium Next GEM Chip H (10x Genomics, 1000161) or the Chromium Single Cell ATAC Library & Gel Bead Kit (10x Genomics, 1000110). Libraries were sequenced on an Illumina NovaSeq 6000 system (1.4 pM loading concentration) using a 50 × 8 × 16 × 49 bp read configuration, targeting an average of 25,000 reads per nucleus.

### Quantification and statistical analysis

**Single-cell methylation data processing (alignment, quality control (QC)).** For alignment and QC of the single-cell methylation data, we used the same mapping strategy used in our previous single-cell methylation projects in our lab[39]. Specifically, we used our in-house mapping pipeline, YAP (https://hq-1.gitbook.io/mc/), for all the mapping-related analysis. The pipeline includes the following main steps: (1) demultiplexing FASTQ files into single cells with Trim Galore (v4.4), (2) reads-level QC, (3) mapping with bismark (v0.20.0), (4) BAM file processing and QC with samtools (1.17) and Picard MarkDuplicates (v3.0.0), and (5) generation of the final molecular profile. Detailed descriptions of these steps for snmC-seq2 can be found in ref. [14]. All the reads were mapped to the human hg38 genome, and we calculated the methylcytosine counts and total cytosine counts for two sets of genomic regions in each cell after mapping.

We filtered out low-quality cells based on three metrics generated during mapping—mapping rate >50%, final mC reads >500,000 and global mCG >0.5. Chromosomes X, Y and M were excluded from the analysis and the remaining genome was divided into 5-kb bins to create a cell-by-bin matrix. In this matrix, each bin was assigned a hypomethylation score (hyposcore) calculated from the $P$ values of a binomial test, which indicates the probability of hypomethylation of that bin. The matrix was further binarized for downstream analysis using a hyposcore cutoff ≥0.95.

Hyposcore measures the likelihood of observing greater than $m$ methylated reads under the assumption that methylation follows the binomial distribution with parameters $c$ and $p$.

$$p = \sum_{i=1}^{n} m_i c_i$$

where $m$ is the observed number of methylated count for region $i$, $c$ is the coverage (total count) covering region $i$, $n$ is the total number of 5-kb bin regions and $p$ is the expected probability of methylation for this cell.

Let's assume

$$P(X = x) = \binom{c}{x} p^x (1-p)^{c-x}$$

then for each 5-kb bin,

$$\text{Hyposcore} = 1 - \sum_{k=0}^{m} \binom{c}{k} p^k (1-p)^{c-k}$$

The calculation of hyposcore was implemented in ALLCools (v1.1.1, https://lhqing.github.io/ALLCools/intro.html) using SciPy[40].

Bins covered by fewer than five cells and those with any absolute $z$ score (the number of cells with nonzero values) >2 were filtered out. Additionally, we excluded bins that overlapped with the ENCODE blacklist using 'bedtools intersect' (Dale, Pedersen and Quinlan 2011; Quinlan and Hall 2010).

**Unsupervised clustering.** To perform unsupervised clustering, we used ALLCools[39], which first conducted PCA on the 5-kb bin matrix. For each exposure, we selected the top 32 principal components (PCs) for clustering using the modules in scanpy. In the HIV-1 and influenza cohorts, we observed a donor effect in the clustering results with these PCs. Therefore, we applied harmony[41] to correct the donor effect on these PCs. We performed clustering separately for control samples ('HIV_pre', 'Flu_pre' and 'Ctrl') and samples from the 'MRSA/MSSA', 'BA', 'COVID-19' and 'OP' groups, allowing for better comparison between the exposures and control samples.

To annotate the cells, we used both the single-cell methylation clustering results and cell-surface markers. In almost every cohort, we observed two clusters of B cells and NK cells, which were distinguished by their global mCG levels. Therefore, we assigned these clusters as naive and memory B cells, naive and active NK cells. We also merged clusters with cell-surface markers indicating memory CD4 and CD8 T cells, even if they exhibited multiple clusters in the t-SNE embedding.

**eDMRs identification.** To identify DMRs associated with each immune cell type, we analyzed PBMCs from healthy donors. Based on single-cell methylation and FACS, we identified nine cell types through clustering. These cell types were grouped based on their global mCG levels, and DMRs were called separately within high-mCG and low-mCG cell types. We used methylpy (v1.4.6, https://github.com/yupenghe/methylpy) for DMR calling and the resulting DMRs were further annotated with genes and promoters.

To identify DMRs associated with each exposure, we merged the control samples and samples from each exposure group. We used methylpy (https://github.com/yupenghe/methylpy) to identify DMRs between the control and exposure groups and between different exposure groups. Once we obtained the primary set of DMRs, we calculated the methylation levels of all samples at these DMRs using 'methylpy add-methylation-level'.

Additional filtering of the DMRs was performed by comparing methylation levels across sample groups using Student's $t$ test. Only DMRs with a minimum $P$ value <0.05 between any two groups were retained. For DMRs associated with MRSA/MSSA, BA, OP and SARS-CoV-2, where external controls were used for DMR calling, we compared the methylation levels of exposure samples and control samples, as well as different cohorts of controls (HIV, Flu and commercial controls). DMRs that showed significant differences ($P < 0.05$)

between the exposure group and all three control cohorts, but no significant differences ($P > 0.05$) between any two control cohorts, were retained.

To visualize complex heatmaps, we used PyComplexHeatmap (https://github.com/DingWB/PyComplexHeatmap)[38,42]. Hypomethylated DMRs in the corresponding sample groups and cell types were labeled for better visualization. The heatmap rows were split according to sample groups and the columns were split based on DMR groups and cell types. Within each subgroup, rows and columns were clustered using Ward linkage and the Jaccard metric.

**Validation of DMRs by shuffling the samples.** To validate that the identified DMRs for each exposure were not confounded by batch effects or other factors, we shuffled the group labels of the samples within each exposure and identified DMRs among the randomly assigned groups. We quantified the methylation levels of all samples at the DMRs from the random groups and performed $t$ tests on the methylation levels between each pair of groups.

**Effect-size calculation.** To quantify the magnitude of methylation differences across exposure groups, we calculated Cohen's $d$ effect sizes for each DMR using pairwise comparisons. For each DMR, methylation values were extracted across the three defined groups. Cohen's $d$ was computed using the pooled standard deviation formula:

$$d = (mean_1 - mean_2)/s\_pooled$$

where $mean_1$ and $mean_2$ are the group means and $s\_pooled$ is the square root of the weighted average of group variances. The pooled variance was calculated with Bessel's correction to account for sample size differences. For each DMR, three pairwise $d$ values were computed—Group1 versus Group2, Group1 versus Group3 and Group2 versus Group3. These effect sizes provide an interpretable measure of methylation divergence independent of sample size or statistical significance.

**SNP calling.** We merged the BAM files from the same donors and SNP calling was performed using Biscuit's[32] variant calling function. This process identifies SNPs in both CpG and nonCpG contexts by analyzing the bisulfite-treated reads. Biscuit distinguishes between methylated cytosines and actual C/T polymorphisms, reducing the risk of false positives. To increase variant density and coverage, we imputed SNPs using Minimac4, referencing the 1,000 Genomes Phase 3 panel. Postimputation, only high-confidence SNPs present in dbSNP were retained to ensure reliability and compatibility with downstream analyses.

Standard variant filtering was applied to remove low-confidence SNPs. We excluded SNPs with a minor allele frequency (MAF) below 0.05. Additionally, SNPs overlapping with regions in the blacklist were filtered out.

**Genetic PCA analysis.** We performed PCA of genotypes following standard best practices to control for population structure as follows:

1.  Linkage disequilibrium (LD) pruning—we used PLINK 2 to filter variants to high-quality, common biallelic SNPs (--snps-only just-a-t, --max-alleles 2, --geno 0.02 and --maf 0.05) and excluded regions of extended LD (high-LD-regions-hg38.bed). We then applied LD pruning with a 200-SNP window, step size of 50 SNPs and an $r^2$ threshold of 0.1 (--indep-pairwise 200 50 0.1).
2.  Preparation of input genotypes—the pruned SNP set was extracted to generate a reduced variant call format (VCF) for PCA, ensuring that only informative and independent variants were retained.

3.  PCA computation:
    (a) The primary analysis was conducted using QTLtool with the options --center, --scale, --maf 0.05 and --distance 0, which ensures mean centering, scaling and consistent handling of pruned SNPs.
    (b) For cross-validation, we also performed PCA with PLINK 2 (--pca 20 approx var-wts), which produced highly concordant results.

This approach ensured robust inference of genetic PCs, minimizing the impact of LD and technical artifacts and provided reliable covariates for downstream QTL and association analyses.

**Identification of gDMR–meQTL pairs.** To identify meQTLs associated with DMRs, we used QTLtools (v2.0-7-g61a04d2c5e)[43]. The analysis was conducted using two approaches—nominal and permutation-based methods—both designed to account for the statistical significance of the association between SNPs and methylation levels. DMRs for each cell type were identified across the 110 donors using methylpy.

**meQTL mapping.** *Nominal analysis.* We used QTLtools in nominal mode to calculate the association between genotype (SNP) and methylation levels within DMRs. This method tests all SNP–DMR pairs within a specified genomic window (1 Mb) around the DMRs, reporting nominal $P$ values for each pair. Associations were considered significant at an FDR threshold <0.01.

*Permutation-based analysis.* To assess the bias in the proportion of COVID-19 samples across different monocyte clusters, a chi-square test was applied. This test helped determine whether the distribution of samples from severe and nonsevere COVID-19 patients differed significantly from that of control samples, with results indicating a strong statistical difference ($P = 2.05 \times 10^{-237}$, Fisher's exact test).

*Trans-meQTL mapping.* We used TensorQTL in *trans* mode to efficiently test large-scale genotype–methylation associations. By default, TensorQTL computes parametric $P$ values using linear regression and applies Benjamini–Hochberg FDR correction.

*Covariates and adjustment.* In both analyses, we included covariates such as age, sex, the first five PCs of genotypes and exposures. Covariate adjustment was performed using the QTLtools built-in method for linear model regression.

**Motif enrichment.** We obtained the hypo- and hyper-DMRs reported by methylpy from the columns 'hypermethylated_samples' and 'hypomethylated_samples'. HOMER (v5.1) was used to identify enriched motifs within these different sets of DMRs for each exposure. The results from HOMER's 'knownResults.txt' output files were used for downstream analysis. Only motif enrichments with a $P$ value <0.01 were retained. Motif enrichment results were visualized using scatter plots generated with Seaborn.

**DMG identification.** Pairwise DMG analysis for each exposure was performed using ALLCools, following the tutorial (https://lhqing.github.io/ALLCools/cell_level/dmg/04-PairwiseDMG.html). Significantly DMGs were selected based on an FDR < 0.01 and a delta mCG > 0.05. The functional enrichment analysis of the DMGs was conducted using Metascape[44] (v3.5, https://metascape.org/). We also used linear regression (mCG ~ annotation + age + sex + ethnicity) to identify the genes associated with the two clusters of monocytes, regressing out age, sex and ethnicity of the donors.

**Integration with single-cell ATAC.** We integrated our single-cell methylation data with single-cell ATAC–seq data from HIV-1. This integration was performed using canonical correlation analysis based on 5-kb bins, where we transferred our methylation cell annotations to the cells from

the other modality. To generate the peaks and BigWig files for each cell type, we used SnapATAC2 (refs. 45,46).

**Correlation of single-cell methylation and single-cell ATAC.** To assess the correlation between single-cell methylation and single-cell ATAC, we calculated the correlation between the hyposcore of each 5-kb bin and Tn5 insertions in each bin. This correlation was performed both across different cell types and within matched cell types.

**Colocalization of meQTL with GWAS traits.** Summary statistics of GWAS were downloaded from the NHGRI-EBI GWAS Catalog (https://www.ebi.ac.uk/gwas/)[47], including 29,401 studies and 25,111 traits. We performed colocalization analysis with coloc[48] (v5.2.3) using default priors to calculate the probability that both the meQTL and GWAS traits share a common causal variant. The posterior probability (PP4) of a single causal variant associated with both DMR and GWAS traits was used to identify significant colocalizations (PP4 > 0.50). A high PP4 value indicates strong evidence for shared causality. R packages locuscomparer (v1.0.0)[49] and locuszoomr (v0.3.1)[50] were used to visualize the colocalization results. To assess whether meQTL and GWAS SNPs were significantly overlapping for each DMR–trait pair, we performed chi-squared tests using the 'stats.chi2_contingency' function from the Python package SciPy[40]. Resulting $P$ values were adjusted for multiple testing using the Benjamini–Hochberg method.

*SMR analysis.* To investigate the relationship of eDMRs and gDMRs on gene expression, we performed SMR (v1.0) analysis[34] by integrating our unfiltered meQTLs with the eQTLs derived from whole-blood gene-expression levels generated from the eQTLGen consortium[34,51].

By setting the exposure as DMR and outcome as gene expression in the SMR analysis, we tested whether genetic variants associated with DMRs are also associated with expression levels of nearby genes, providing evidence for associations between the two. We did not conduct HEIDI analysis due to the differences between the two sample populations. This does not rule out the possibility that some of the DMR–gene associations identified in our SMR analysis are due to linkage rather than pleiotropy or causality. SMR associations with a $P$ value below the Bonferroni-adjusted alpha level ($\alpha = 0.05/n$) were considered significant. Default parameters were used to run SMR. We further filtered out associations in the HLA region (chr6: 28,477,797–33,448,354).

**Statistics.** *Enrichment tests.* Enrichment tests were conducted using Fisher's exact test to evaluate the distribution of DMRs across exposures and cell types. This statistical approach was selected due to the small sample sizes in some groups and the need for exact calculations without relying on large-sample approximations.

*Bias analysis of COVID samples in monocyte clusters.* To assess the bias in the proportion of COVID-19 samples across different monocyte clusters, a chi-square test was applied. This test helped determine whether the distribution of samples from severe and nonsevere COVID-19 patients differed significantly from that of control samples, with results indicating a strong statistical difference ($P = 2.05 \times 10^{-237}$, Fisher's exact test).

*Effect-size calculation.* For each DMR, effect sizes were calculated using Cohen's $d$ to quantify the magnitude of methylation differences among exposure groups. This measure allows for an understanding of the biological relevance of the observed methylation changes, independent of sample size. Cohen's $d$ was computed for pairwise comparisons across groups (for example, exposure versus control), using the pooled standard deviation formula.

*Differential methylation analysis.* We performed pairwise differential methylation analysis between exposure groups using the methylpy

package. Statistical significance was determined by calculating $P$ values for each comparison, with a threshold of 0.05. Further, we applied FDR correction to adjust for multiple comparisons.

### Ethics

The study was conducted with approval from the Salk Institutional Review Board (IRB) under protocol 18-0015 titled 'Single Cell Analysis for Forensic Epigenetics (SAFE)'. Research activities were covered under Salk's Federalwide Assurance (FWA) for the Protection of Human Subjects (FWA00005316).

### Reporting summary

Further information on research design is available in the Nature Portfolio Reporting Summary linked to this article.

### Data availability

De-identified molecular data and associated sample metadata generated in this study are available through controlled access via the Database of Genotypes and Phenotypes (dbGaP) under accession phs003204.v1.p1 (https://dbgap.ncbi.nlm.nih.gov/beta/study/phs003204.v1.p1/). Access to these data is subject to approval by the dbGaP Data Access Committee in accordance with NIH policies on the sharing of human genomic data. The single-cell ATAC–seq data were generated in our companion study[52] and the processed data used in this study have been deposited in the NCBI Gene Expression Omnibus (GEO) under accession GSE306525 (https://www.ncbi.nlm.nih.gov/geo/query/acc.cgi?acc=GSE306525). All other data supporting the findings of this study are available within the article and its Supplementary Information files. Source data are provided with this paper.

### Code availability

Codes of all the analyses are available on GitHub (https://github.com/wangwl/ECHO) and Zenodo (https://doi.org/10.5281/zenodo.17307293) (ref. 53).

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

## Acknowledgements

This work was supported by the Defense Advanced Research Projects Agency (DARPA) through the DARPA Epigenetic Characterization and Observation (ECHO) program for the project Single-cell Analysis for Forensic Epigenetics (SAFE), administered by the US Army Research Office under cooperative agreement W911NF-19-2-0185. J.R.E. is an investigator of the Howard Hughes Medical Institute. We thank former DARPA Biological Technologies Office Program Manager E.V. Gieson, as well as the current leadership team, including J.-P. Chretien and T. Thomou, for their guidance and insightful comments. W.J.G. acknowledges funding support from the National Institutes of Health (NIH; grants P50-HG007735, UM1-HG009442 and UM1-HG009436). S.C.S. acknowledges funding support from DARPA (grant N6600119C4022). V.G.F. acknowledges funding from NIH (grant 1R01AI165671). R.W. was supported in part by the National Institute of General Medical Sciences Predoctoral Basic Biomedical Sciences Research Training Program T32 GM145427. S.C.S. also acknowledges X. Yu, T.E. acknowledges A. Donabedian for sample collection. We extend our gratitude to the ECHO team of Arizona State University, particularly J. LaBaer and V. Murugan, for coordinating the various teams during the initial phase of the study. We thank all anonymous donors who contributed biological samples to this project through our collaborators. We thank S. Kaech for consultations on the PBMC selection strategy, and C. O'Connor and L. Boggeman at the Salk FACS Core for their assistance in standardizing the gating strategy. This work used the Stampede2 supercomputing resources at Texas Advanced Computing Center (TACC) through the Extreme Science and Engineering Discovery Environment (XSEDE) and the Anvil supercomputing resources at Purdue University's Rosen Center for Advanced Computing (RCAC), made available through the Advanced Cyberinfrastructure Coordination Ecosystem: Services & Support (ACCESS) program. These resources were funded by the National Science Foundation (NSF) under grants 1540931 and 2005632, respectively, and were assessed via a research allocation to the Ecker Lab (grant MCB130189). We thank M. Gujral, L. Huang and R. Scott, and other engineers at TACC and RCAC, for their assistance in porting and optimizing computational tools on XSEDE resources, provided through the XSEDE Extended Collaborative Support Service (ECSS) program. We thank K. Claypool, R. Jaimes, A. Michaleas, D. Ricke, P. Fremont-Smith and T. White from MIT Lincoln Laboratory for providing data warehousing support. Some items in Fig. 1 were created with BioRender.com.

## Author contributions

J.R.E. and M.H. conceived the study. J.R.E. and W.J.G. supervised the study. W.W., M.H., W.D., R.W. and V.R. performed data analysis. A.B., C.B., R.C., A.A., J.A., M.K. and J.R.N. performed snmC-seq2, with A.B., C.B. and R.C. also involved in standardizing the PBMC gating strategy and FACS. H.S., V.R., W.D., H.L., W.T., J.Z., B.W., F.M., Q.Z. and I.B.G. were involved in data analysis. H.C. and J.R.N. performed data management. M.T.M., L.L.S., T.J.B., E.A.P., X.S., C.W.W., V.G.F., T.N., M.T.M., P.P., D.B.B. and F.R. collaborated in the selection of the PBMC samples for the HIV, MRSA, MSSA, COVID and OP cohorts. J.L.B., A.K.S. and R.R.S. collaborated in the selection of the PBMC samples for the Anthrax cohort. S.V., I.R., G.N., S.C.S., F.C., A.M.W. and T.E. collaborated in the selection of the PBMC samples for the Flu cohort. W.W. designed the research framework, did the investigation and wrote the manuscript. All authors read the manuscript and provided comments.

## Competing interests

An application for a patent based on the results of this work has been filed with the United States Patent and Trademark Office (USPTO) under application US 63/489,546. J.R.E. is a scientific advisor for Zymo Research Inc. and Ionis Pharmaceuticals. W.J.G. is named as an inventor on patents describing ATAC–seq methods. 10x Genomics has licensed intellectual property on which W.J.G. is listed as an inventor. W.J.G. holds options in 10x Genomics and is a consultant for Ultima Genomics and Guardant Health. W.J.G. is a scientific cofounder of Protillion Biosciences. V.G.F. reports personal fees from Novartis, Debiopharm, Genentech, Achaogen, Affinium, Medicines, MedImmune, Bayer, Basilea, Affinergy, Janssen, Contrafect, Regeneron, Destiny, Amphliphi Biosciences, Integrated Biotherapeutics, C3J, Armata, Valanbio, Akagera and Aridis, Roche; grants from NIH, MedImmune, Allergan, Pfizer, Advanced Liquid Logics, Theravance, Novartis, Merck, Medical Biosurfaces, Locus, Affinergy, Contrafect, Karius, Genentech, Regeneron, Deep Blue, Basilea and Janssen; royalties from UpToDate; stock options from Valanbio and ArcBio; honoraria from Infectious Diseases of America for service as Associate Editor of Clinical Infectious Diseases; and a patent for sepsis diagnostics pending. S.C.S. is the scientific founder and serves as Chief Scientific Officer of GNOMX. This article was prepared while I.R. was employed at the Icahn School of Medicine at Mount Sinai. The opinions expressed in this article are the authors' own and do not reflect the view of the National Institutes of Health, the Department of Health and Human Services or the United States government. The other authors declare no competing interests.

## Additional information

**Extended data** is available for this paper at https://doi.org/10.1038/s41588-025-02479-6.

**Correspondence and requests for materials** should be addressed to Joseph R. Ecker.

## a. Gating strategy

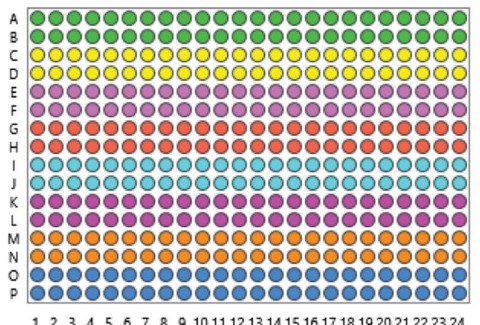

### b. Gates and Statistics from one FACS result

| Name | Events | %Parent | %Total |
|---|---|---|---|
| ☐ All Events | 100,000 | 0.00% | 100.00% |
| ☐ Live | 93,918 | 93.92% | 93.92% |
| ☐ Scatter | 92,860 | 98.87% | 92.86% |
| ☐ Singlets | 79,500 | 85.61% | 79.50% |
| ☐ CD3+ | 48,455 | 60.95% | 48.46% |
| ☐ CD8+ | 13,144 | 27.13% | 13.14% |
| ☐ CD8 Memory | 4,205 | 31.99% | 4.21% |
| ☐ CD8 Naive | 4,444 | 33.81% | 4.44% |
| ☐ CD4+ | 32,377 | 66.82% | 32.38% |
| ☐ CD4 Naive | 13,056 | 40.32% | 13.06% |
| ☐ CD4 Memory | 15,158 | 46.82% | 15.16% |
| ☐ CD3- | 25,914 | 32.60% | 25.91% |
| ☐ CD19- | 20,646 | 79.67% | 20.65% |
| ☐ CD14- | 16,736 | 81.06% | 16.74% |
| ☐ Other (--) | 13,127 | 78.44% | 13.13% |
| ☐ NK | 2,159 | 12.90% | 2.16% |
| ☐ CD14+ | 1,394 | 6.75% | 1.39% |
| ☐ CD19+ | 4,858 | 18.75% | 4.86% |

### c. Plate strategy

| Sort ID | Sort Gate | Color | Sort Mode | Cell Size | Stop Count | Timeout |
|---|---|---|---|---|---|---|
| Sort ID 1 | CD8 Memory | | Single Cell | Regular Cell | 1 | 0 |
| Sort ID 2 | NK | | Single Cell | Regular Cell | 1 | 0 |
| Sort ID 3 | CD14+ | | Single Cell | Regular Cell | 1 | 0 |
| Sort ID 4 | CD19+ | | Single Cell | Regular Cell | 1 | 0 |
| Sort ID 5 | CD4 Memory | | Single Cell | Regular Cell | 1 | 0 |
| Sort ID 6 | CD4 Naive | | Single Cell | Regular Cell | 1 | 0 |
| Sort ID 7 | CD8 Memory | | Single Cell | Regular Cell | 1 | 0 |
| Sort ID 8 | CD8 Naive | | Single Cell | Regular Cell | 1 | 0 |

**Extended Data Fig. 1 | FACS gating process and plate pooling strategy. a**, An example gating process for one sample. **b**, An example of gating statistics of one sample. **c**, We sorted different cell types in the same plate for each sample.

**a**

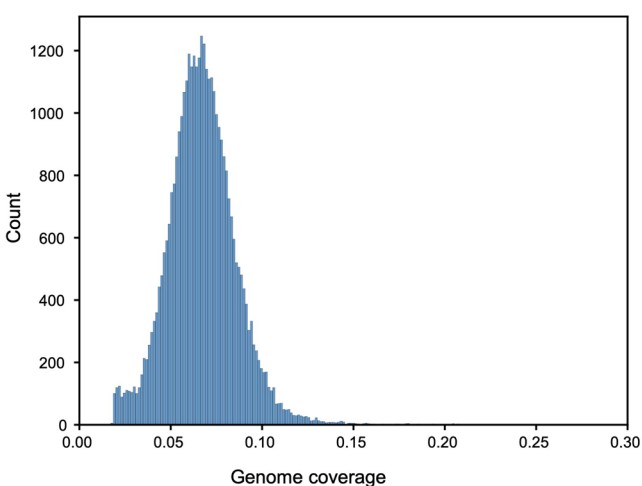

**b**

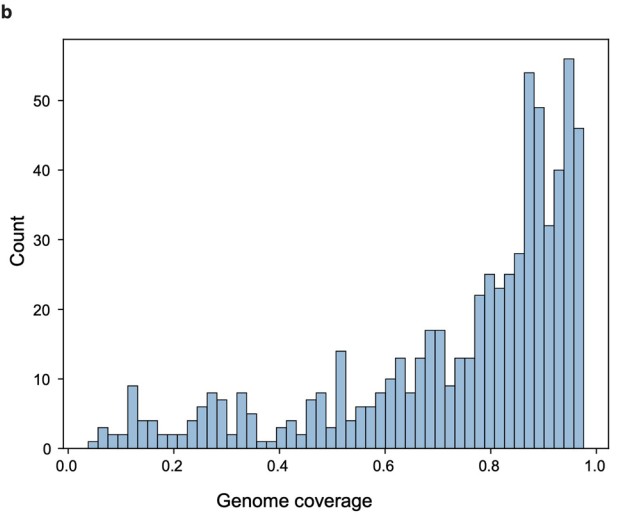

**c**

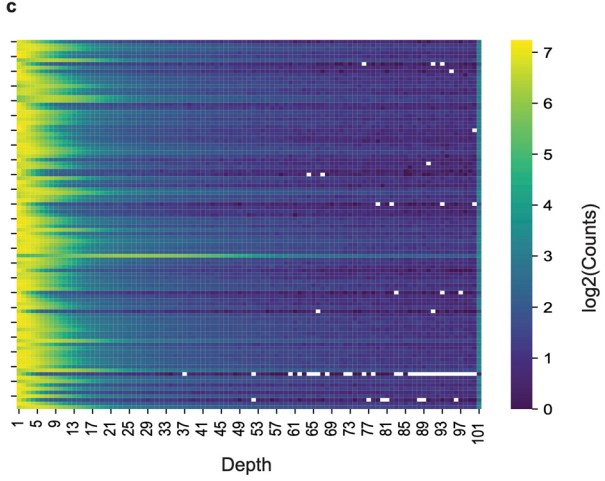

**d**

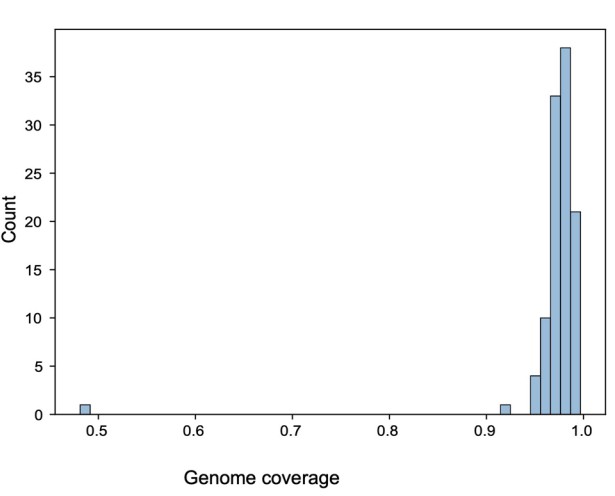

**e**

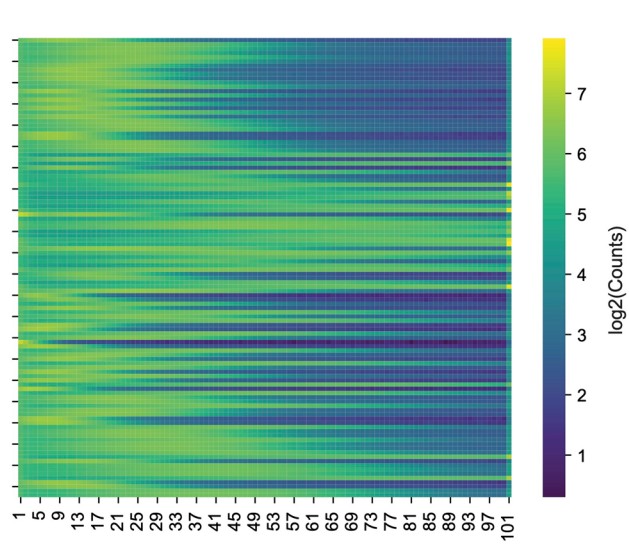

**Extended Data Fig. 2 | See next page for caption.**

**Extended Data Fig. 2 | Quality control of the single-nucleus methylation sequencing data and merged pseudobulk for each sample and condition.** **a**, The genome coverage distribution of all single cells sequenced in this study. **b**, The CpG sites coverage distribution of each merged sample. **c**, Heatmap shows the distribution of sequencing depth at all CpG sites of all merged samples, each row is a sample, x-axis shows the covered depth at CpG site, color shows the number of CpG sites at that depth. **d**, The CpG sites coverage distribution of all cells merged in each exposure condition. **e**, Heatmap shows the distribution of sequencing depth at all CpG sites of all merged exposure conditions, each row is a condition, x-axis shows the covered depth at CpG site, color shows the number of CpG sites at that depth.

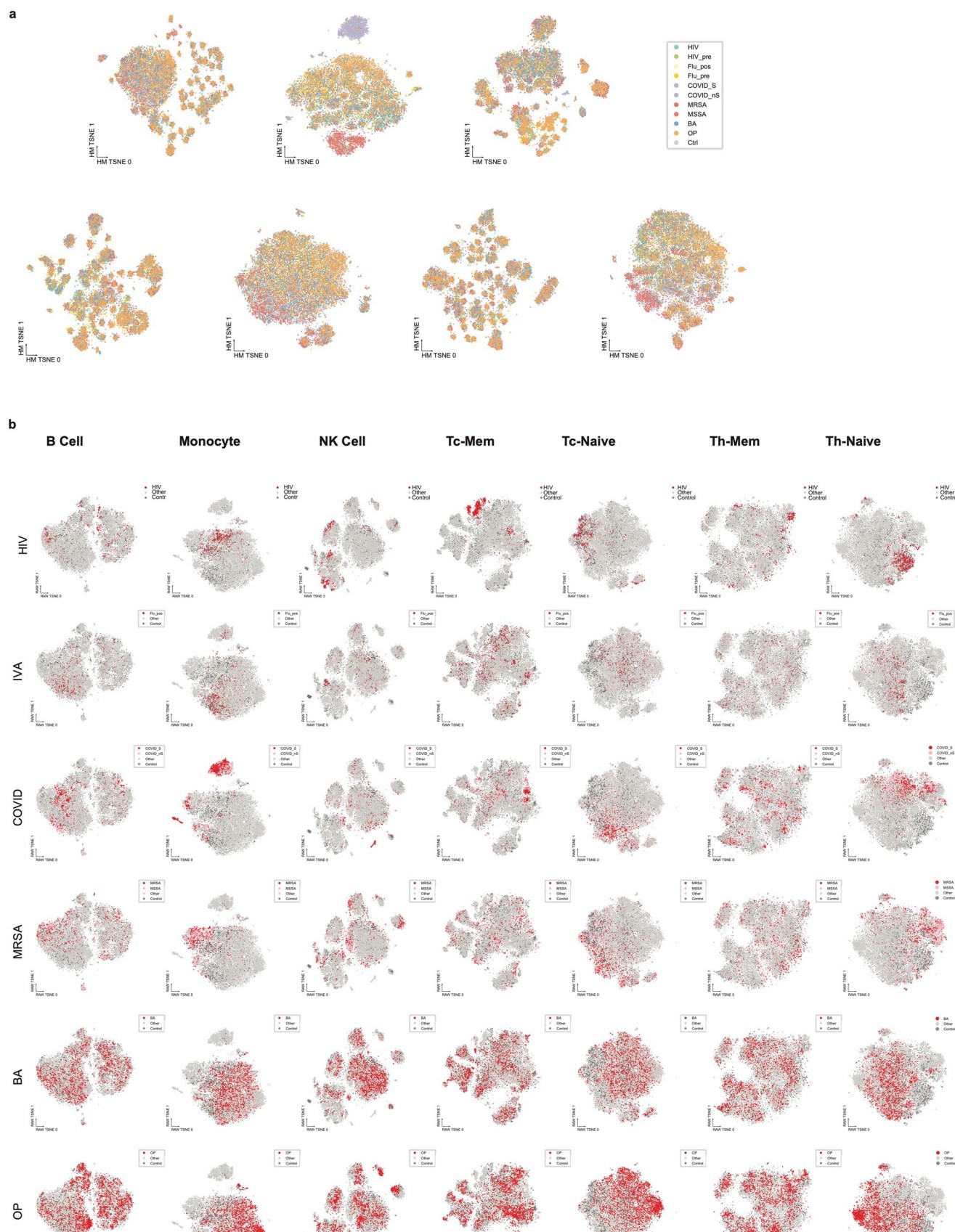

**Extended Data Fig. 3 | Distribution of cells from each exposure within-cell-type t-SNE. a**, t-SNE after harmony integration of each sorted cell type, colored by exposures. **b**, The cells from the corresponding exposure are colored in red, while other cells and control cells are colored in dark gray and gray.

**a**

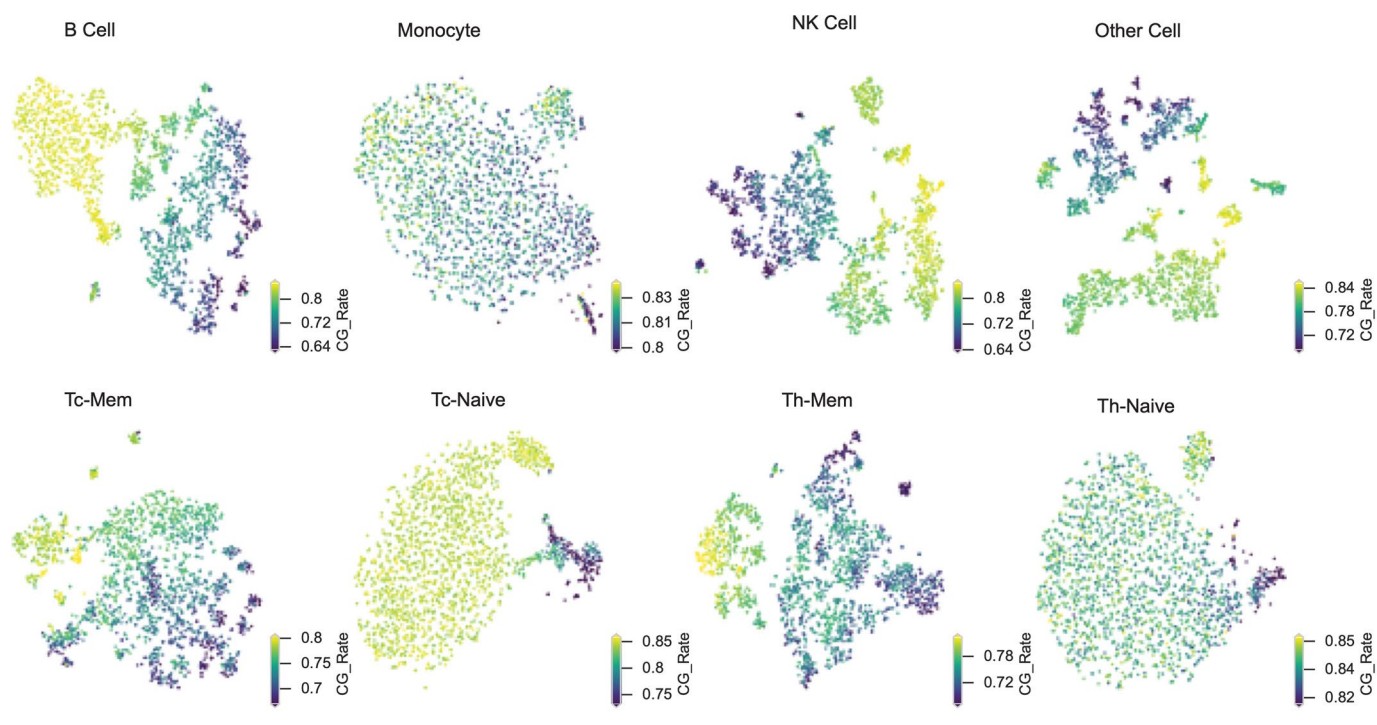

**b**

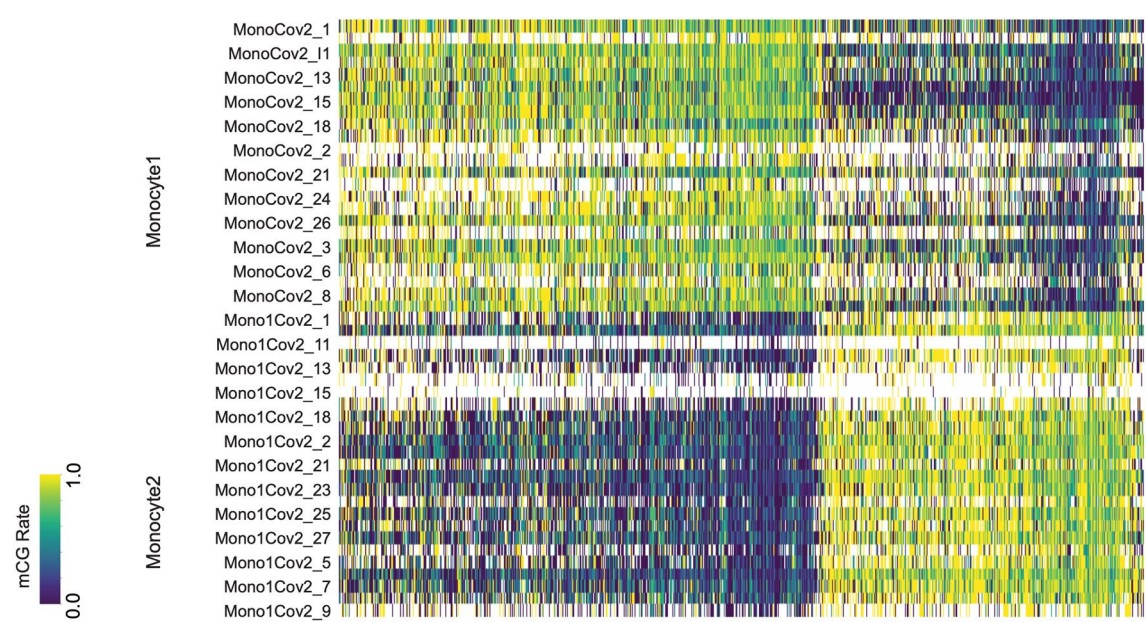

**Extended Data Fig. 4 | Within-cell-type methylation difference between sub-clusters. a**, The UMAP of cells from HIV exposure in each FACS cell type. The color shows the global methylation level of each cell. **b**, Methylation level of DMGs between the two clusters of monocytes in COVID samples.

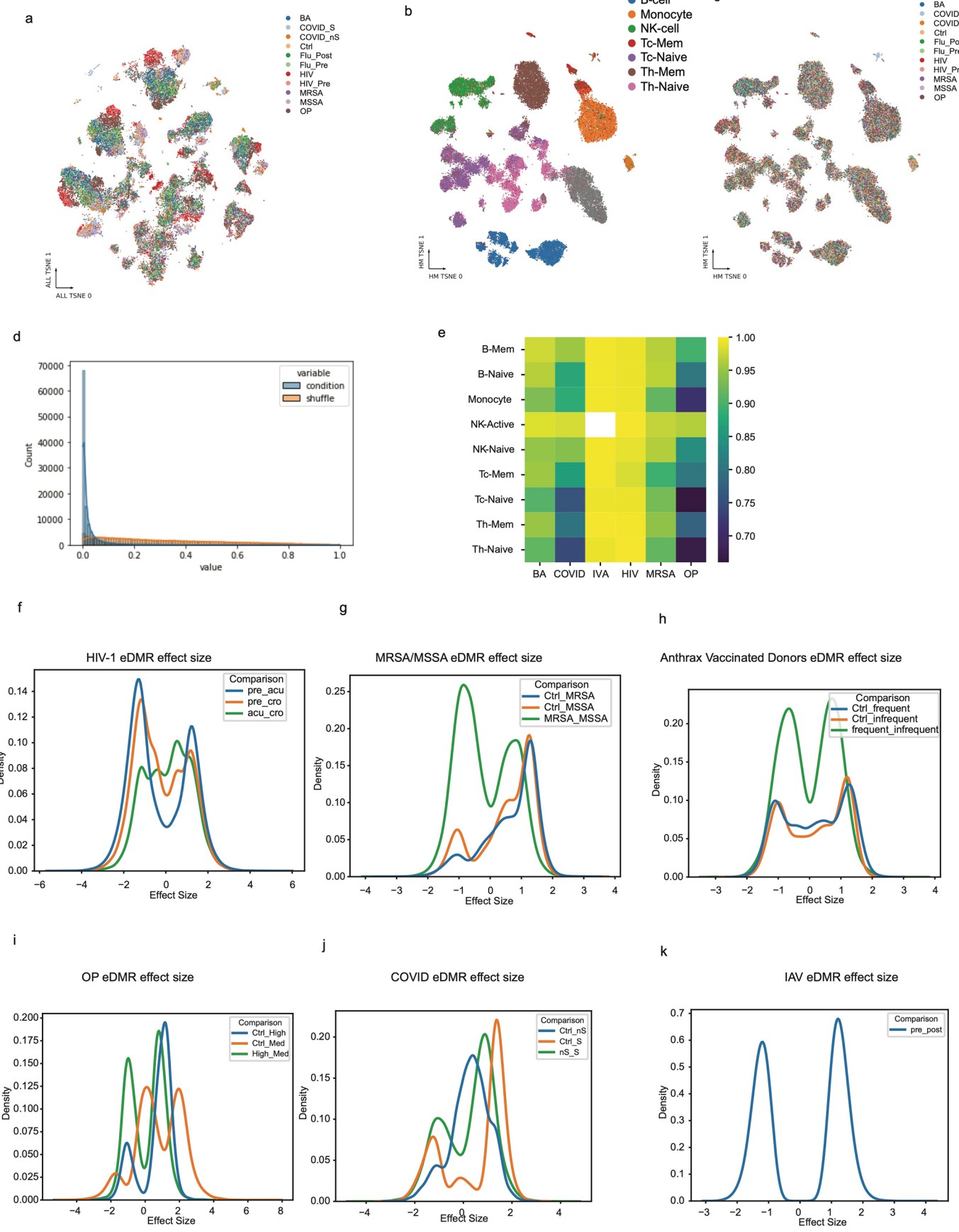

**Extended Data Fig. 5 | Quality control of eDMR calling. a**, UMAP without harmony integration, colored by exposures. **b**, UMAP after harmony integration by donors, colors show the cell types by FACS sorting. **c**, UMAP after harmony integration by donors, colors show the exposures. **d**, Histplot shows the distribution of p values when calling exposure conditions eDMRs and DMRs from label shuffled samples. **e**, Heatmap shows the ratio of single CpG eDMRs in each exposure and cell type. **f–k**, Plots show the effect size distributions of eDMRs from each exposure.

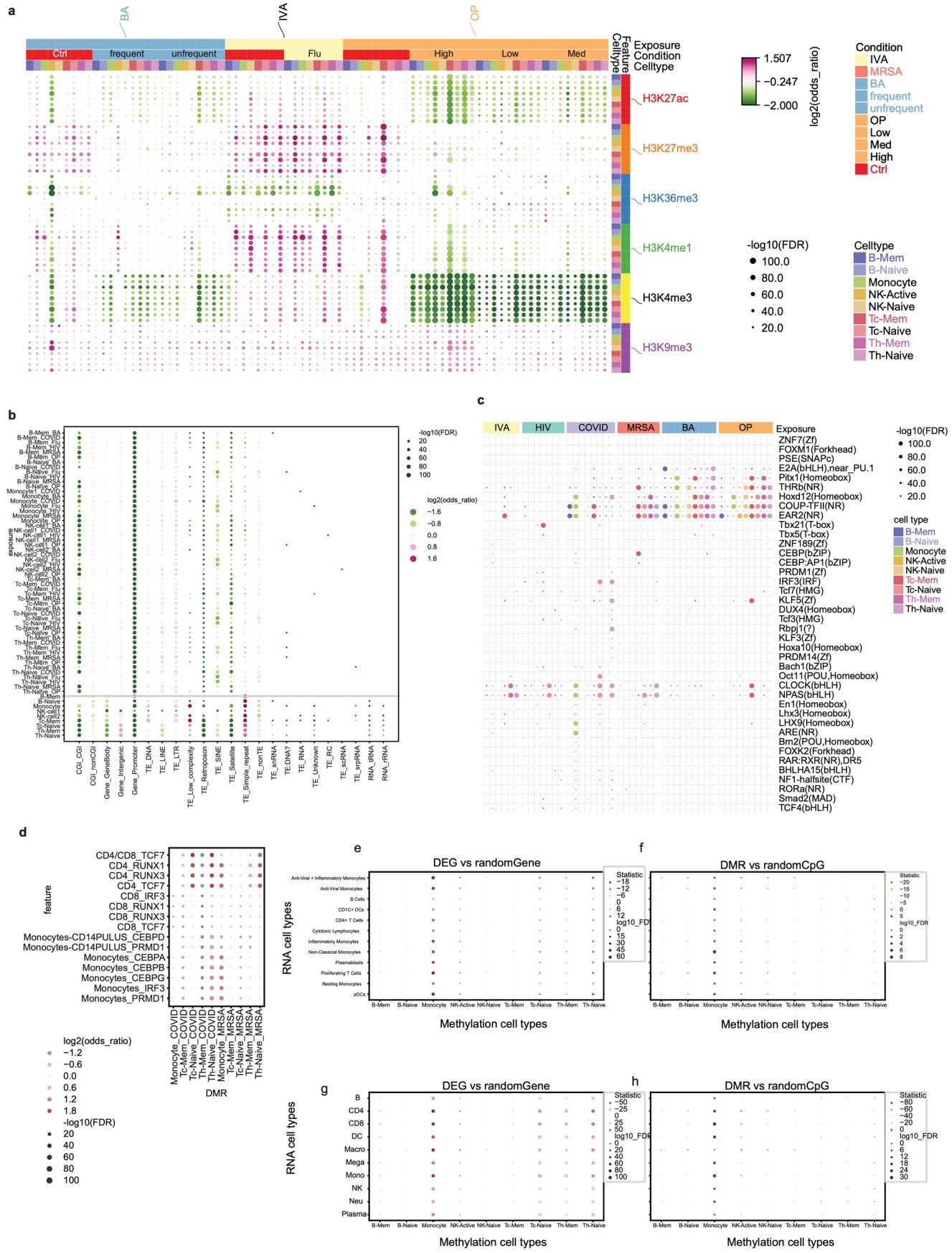

**Extended Data Fig. 6 | See next page for caption.**

**Extended Data Fig. 6 | Features of eDMRs. a**, Dot plot shows the enrichment of eDMRs from BA, influenza virus and OP in histone modification peaks. Each column shows the hypo-eDMRs in that condition. Color of the dots shows the enrichment or depletion in the corresponding histone modification.
**b**, Genomic features enrichment of eDMRs in each cell type and exposure.
**c**, Motif enrichment of hypo-eDMRs in each cell type and exposure.

**d**, Dot plots show the enrichment of eDMRs from each cell type in COVID-19 and MRSA/MSSA with the transcription factor ChIP-seq peaks in the corresponding cell types. **e,f**, Dot plots show the HIV-1-associated eDMRs near DEGs, compared with random genes (**e**) or with random CpG sites (**f**). **g,h**, Dot plots show the COVID-19-associated eDMRs near DEGs, compared with random genes (**g**) or with random CpG sites (**h**).

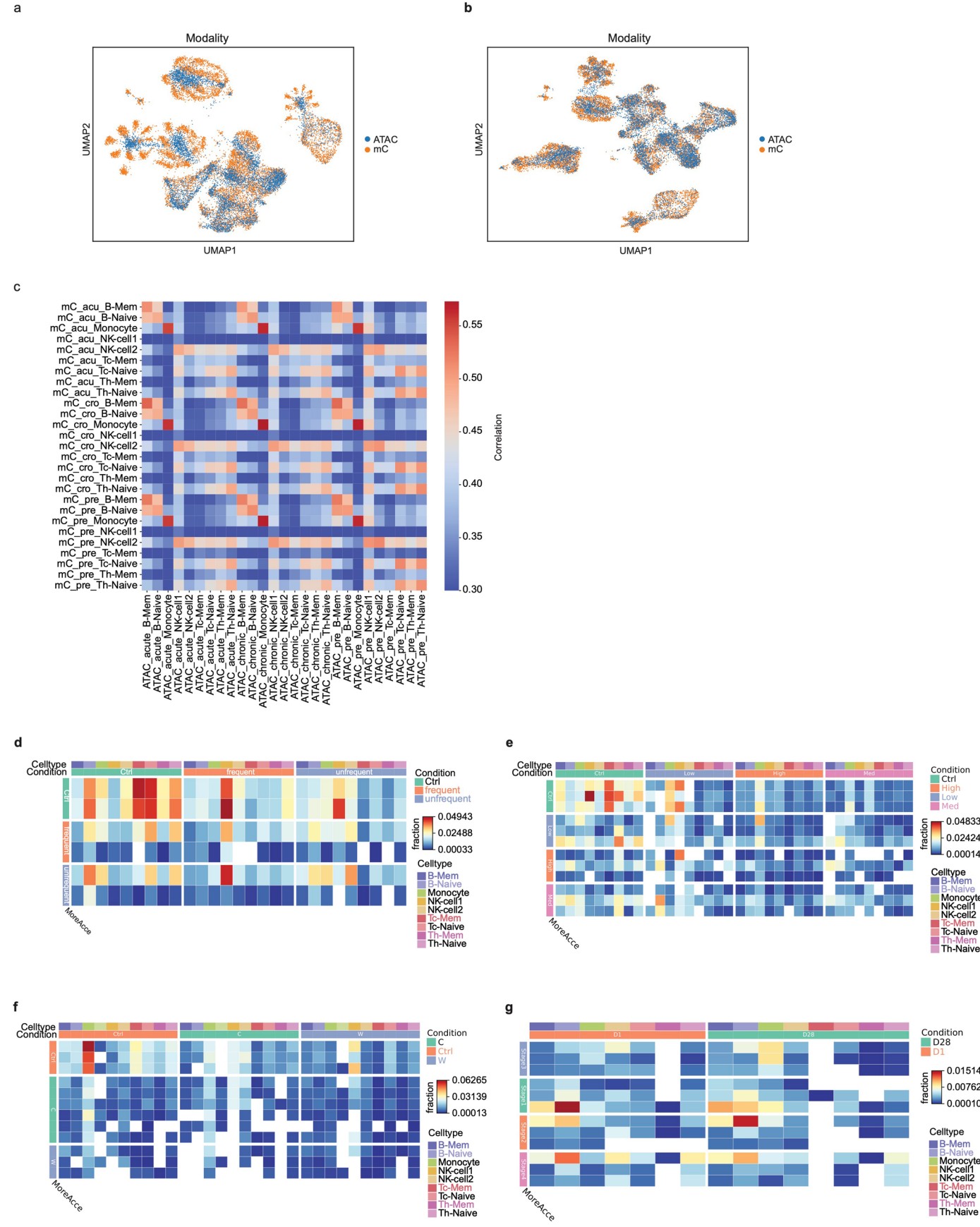

**Extended Data Fig. 7 | See next page for caption.**

**Extended Data Fig. 7 | Integration of snATAC-seq and snmC-seq2. a**, UMAP shows the joint embedding of snATAC-seq and snmC-seq2 data from one sample before harmony integration. **b**, UMAP shows the joint embedding after harmony integration. **c**, Global correlation of DNA methylation and chromatin accessibility using 5 kb bins across the genome. **d**, Heatmap shows the corresponding changes in both DNA methylation and chromatin accessibility in anthrax vaccine exposure. **e**, Heatmap shows the corresponding changes in both DNA methylation and chromatin accessibility in OP exposure. **f**, Heatmap shows the corresponding changes in both DNA methylation and chromatin accessibility in SARS-CoV-2 exposure. **g**, Heatmap shows the corresponding changes in both DNA methylation and chromatin accessibility in IAV exposure.

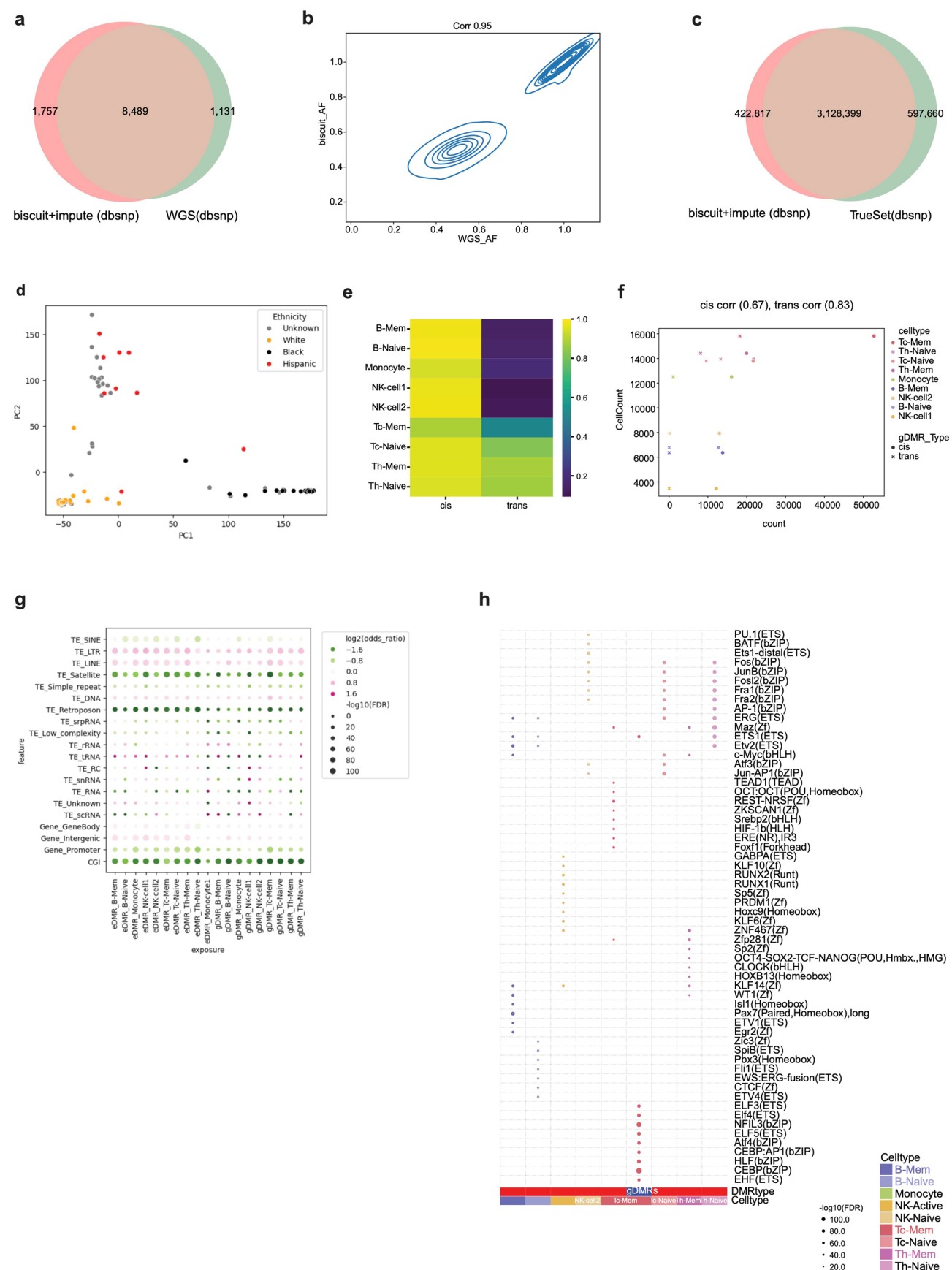

**Extended Data Fig. 8 | See next page for caption.**

**Extended Data Fig. 8 | Features of gDMRs. a**, Venn diagram shows the overlap of SNPs called from methylation reads and WGS in a 10 Mb region. The SNPs are intersected with dbsnp. **b**, kdeplot shows the correlation of alternative allele frequency of the SNPs from WGS and biscuit **c**, Venn diagram shows the overlap of SNPs called from methylation reads and ground truth SNPs for NA12878. The SNPs are intersected with dbsnp. **d**, Scatter plot shows the PCA result of SNPs, the first two PCs were shown, color shows the ethnicity of the donors. **e**, Heatmap shows the ratio of single CpG gDMRs in each cell type in trans and cis. **f**, Scatter plot shows the correlation between gDMR (cis and trans) counts with the number of cells sequenced in each cell type. **g**, Genomic features enrichment of eDMRs and gDMRs. **h**, Motif enrichment of gDMRs and eDMRs using each other as background.

a

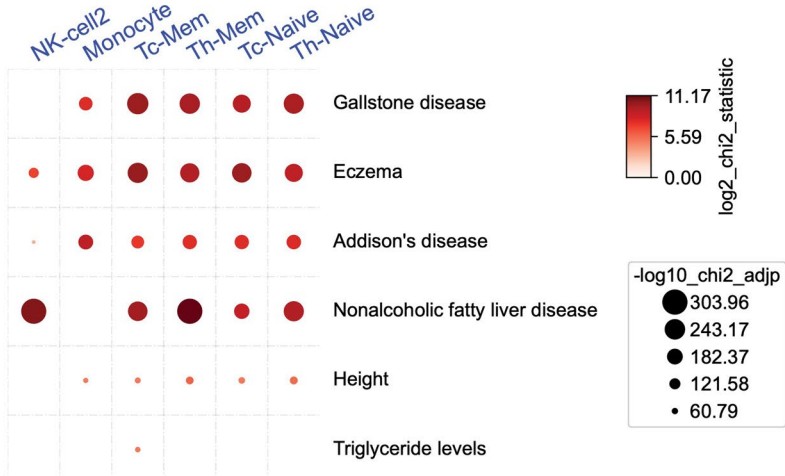

b

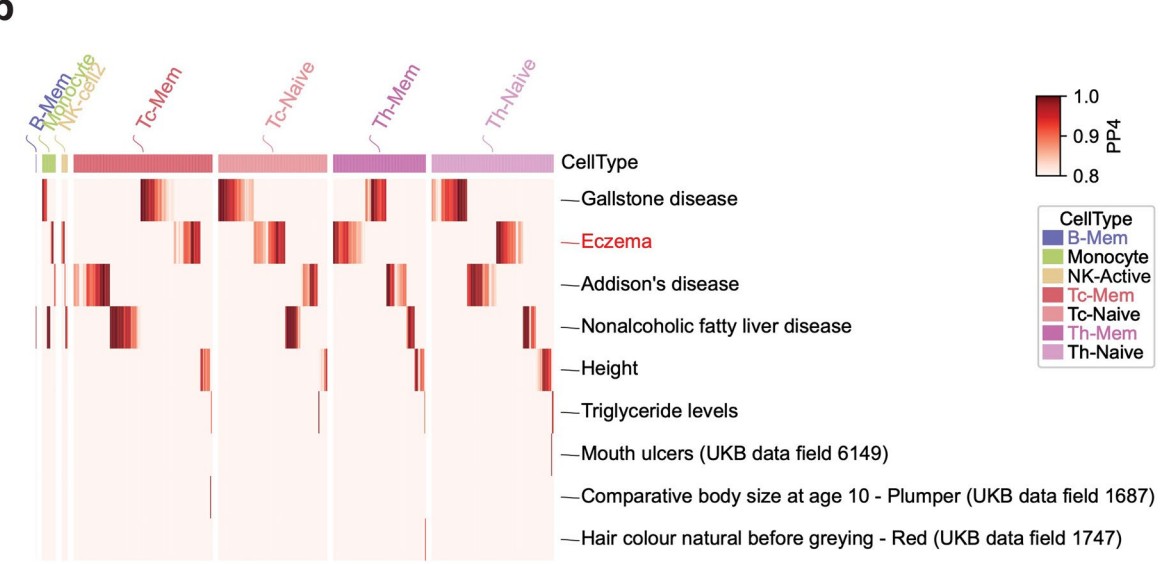

**Extended Data Fig. 9 | Colocalization of meQTL and phenotype-associated GWAS SNPs. a**, Enrichment of colocalized GWAS SNPs from each phenotype with the meQTLs from each cell type. **b**, Heatmap shows the distribution of colocalized meQTLs with different phenotypes in each cell type.

# Reporting Summary

## Statistics

For all statistical analyses, confirm that the following items are present in the figure legend, table legend, main text, or Methods section.

| n/a | Confirmed | |
|---|---|---|
| ☐ | ☒ | The exact sample size (*n*) for each experimental group/condition, given as a discrete number and unit of measurement |
| ☐ | ☒ | A statement on whether measurements were taken from distinct samples or whether the same sample was measured repeatedly |
| ☐ | ☒ | The statistical test(s) used AND whether they are one- or two-sided <br> *Only common tests should be described solely by name; describe more complex techniques in the Methods section.* |
| ☐ | ☒ | A description of all covariates tested |
| ☐ | ☒ | A description of any assumptions or corrections, such as tests of normality and adjustment for multiple comparisons |
| ☐ | ☒ | A full description of the statistical parameters including central tendency (e.g. means) or other basic estimates (e.g. regression coefficient) AND variation (e.g. standard deviation) or associated estimates of uncertainty (e.g. confidence intervals) |
| ☐ | ☒ | For null hypothesis testing, the test statistic (e.g. *F*, *t*, *r*) with confidence intervals, effect sizes, degrees of freedom and *P* value noted <br> *Give P values as exact values whenever suitable.* |
| ☒ | ☐ | For Bayesian analysis, information on the choice of priors and Markov chain Monte Carlo settings |
| ☒ | ☐ | For hierarchical and complex designs, identification of the appropriate level for tests and full reporting of outcomes |
| ☐ | ☒ | Estimates of effect sizes (e.g. Cohen's *d*, Pearson's *r*), indicating how they were calculated |

*Our web collection on statistics for biologists contains articles on many of the points above.*

## Software and code

Policy information about availability of computer code

| Data collection | Codes of all the analysis are available on github (https://github.com/wangwl/ECHO) and Zenodo (10.5281/zenodo.17307293). |
|---|---|
| Data analysis | Software and algorithms <br> bismark (v0.20.0) (Krueger and Andrews, 2011) https://github.com/FelixKrueger/Bismark <br> Trim Galore (4.4) https://www.bioinformatics.babraham.ac.uk/projects/trim_galore/; RRID:SCR_011847 <br> samtools (Li et al., 2009) http://www.htslib.org/ <br> subread (Liao et al., 2013) https://subread.sourceforge.net/ <br> Picard MarkDuplicates (Version:3.0.0) https://broadinstitute.github.io/picard/ <br> ALLCools (1.1.1) https://github.com/lhqing/ALLCools <br> snapATAC2 (2.9.0.dev0) (Fang et al., 2021; <br> Zhang et al., 2021) <br> https://github.com/kaizhang/SnapATAC2 <br> methylpy (1.4.6) (Schultz et al., 2016) <br> https://github.com/yupenghe/methylpy <br> GREAT (McLean et al., 2010; <br> Tanigawa et al., 2022) http://great.stanford.edu/public/html/index.php <br> R (4.3.1) https://cran.r-project.org; RRID:SCR_001905 <br> MACS3 (3.0.3b) (Zhang et al., 2008) https://github.com/taoliu/MACS <br> Homer (v5.1) (Heinz et al., 2010) http://homer.ucsd.edu/homer; RRID:SCR_010881 <br> bedtools (v2.31.1) https://bedtools.readthedocs.io/en/latest/#; RRID:SCR_006646 <br> wigToBigWig (v2.9) http://hgdownload.cse.ucsc.edu/admin/exe/ |

Metascape (v3.5) (Zhou et al., 2019) https://metascape.org/gp/index.html#/main/step1; RRID:SCR_016620
QTLtools (v2.0-7-g61a04d2c5e)  https://qtltools.github.io/qtltools/
coloc (v5.2.3) https://cran.r-project.org/web/packages/coloc/index.html
smr (v1.0)https://yanglab.westlake.edu.cn/software/smr/#Overview

For manuscripts utilizing custom algorithms or software that are central to the research but not yet described in published literature, software must be made available to editors and reviewers. We strongly encourage code deposition in a community repository (e.g. GitHub). See the Nature Portfolio guidelines for submitting code & software for further information.

## Data

Policy information about availability of data

All manuscripts must include a data availability statement. This statement should provide the following information, where applicable:
- Accession codes, unique identifiers, or web links for publicly available datasets
- A description of any restrictions on data availability
- For clinical datasets or third party data, please ensure that the statement adheres to our policy

De-identified molecular data and associated sample metadata generated in this study are available through controlled access via the Database of Genotypes and Phenotypes (dbGaP) under accession number phs003204.v1.p1. Access to these data is subject to approval by the dbGaP Data Access Committee in accordance with NIH policies on the sharing of human genomic data.
The single-cell ATAC-seq data generated and analyzed in this study have been deposited in the NCBI Gene Expression Omnibus (GEO) under accession number GSE306525.
All other data supporting the findings of this study are available within the article and its Supplementary Information files. Source data are provided with this paper.

## Research involving human participants, their data, or biological material

Policy information about studies with human participants or human data. See also policy information about sex, gender (identity/presentation), and sexual orientation and race, ethnicity and racism.

| | |
|---|---|
| Reporting on sex and gender | We have included both male and female sexes in our analysis wherever the samples were available for the various exposures. |
| Reporting on race, ethnicity, or other socially relevant groupings | Self-reported ethnicity, age, sex etc where available for most of the exposure samples we have analyzed. This is provided as Supplementary information. |
| Population characteristics | For most exposures, the donors are from a broad and mixed population. |
| Recruitment | We did not recruit any volunteers for the exposure samples. These were provided by sample provider collaborator sites through their respective separate studies. |
| Ethics oversight | The work was conducted after Salk Institutional Review Board (IRB) approval through IRB Protocol Number: 18-0015 titled "Single Cell Analysis for Forensic Epigenetics (SAFE)." Salk Federal Wide Assurance (FWA) for the Protection of Human Subject Number: FWA00005316. |

Note that full information on the approval of the study protocol must also be provided in the manuscript.

# Field-specific reporting

Please select the one below that is the best fit for your research. If you are not sure, read the appropriate sections before making your selection.

☒ Life sciences          ☐ Behavioural & social sciences          ☐ Ecological, evolutionary & environmental sciences

For a reference copy of the document with all sections, see nature.com/documents/nr-reporting-summary-flat.pdf

# Life sciences study design

All studies must disclose on these points even when the disclosure is negative.

| | |
|---|---|
| Sample size | Previous studies performed in our lab and others have shown that methylation differences using chi-square based tests between cases and control samples can provide signficance levels of at least 0.05 |
| Data exclusions | No data is excluded |
| Replication | Biological replicates including both sexes were used. |
| Randomization | Randomization details are available for each exposure, respectively. |
| Blinding | Investigators have access only to alphanumerical IDs linked to basic metadata. |

# Reporting for specific materials, systems and methods

We require information from authors about some types of materials, experimental systems and methods used in many studies. Here, indicate whether each material, system or method listed is relevant to your study. If you are not sure if a list item applies to your research, read the appropriate section before selecting a response.

## Materials & experimental systems

| n/a | Involved in the study |
|-----|-----------------------|
| ☒ ☐ | Antibodies |
| ☒ ☐ | Eukaryotic cell lines |
| ☒ ☐ | Palaeontology and archaeology |
| ☒ ☐ | Animals and other organisms |
| ☒ ☐ | Clinical data |
| ☒ ☐ | Dual use research of concern |
| ☒ ☐ | Plants |

## Methods

| n/a | Involved in the study |
|-----|-----------------------|
| ☒ ☐ | ChIP-seq |
| ☐ ☒ | Flow cytometry |
| ☒ ☐ | MRI-based neuroimaging |

## Plants

| | |
|---|---|
| Seed stocks | *Report on the source of all seed stocks or other plant material used. If applicable, state the seed stock centre and catalogue number. If plant specimens were collected from the field, describe the collection location, date and sampling procedures.* |
| Novel plant genotypes | *Describe the methods by which all novel plant genotypes were produced. This includes those generated by transgenic approaches, gene editing, chemical/radiation-based mutagenesis and hybridization. For transgenic lines, describe the transformation method, the number of independent lines analyzed and the generation upon which experiments were performed. For gene-edited lines, describe the editor used, the endogenous sequence targeted for editing, the targeting guide RNA sequence (if applicable) and how the editor was applied.* |
| Authentication | *Describe any authentication procedures for each seed stock used or novel genotype generated. Describe any experiments used to assess the effect of a mutation and, where applicable, how potential secondary effects (e.g. second site T-DNA insertions, mosiacism, off-target gene editing) were examined.* |

## Flow Cytometry

### Plots

Confirm that:

☒ The axis labels state the marker and fluorochrome used (e.g. CD4-FITC).

☒ The axis scales are clearly visible. Include numbers along axes only for bottom left plot of group (a 'group' is an analysis of identical markers).

☒ All plots are contour plots with outliers or pseudocolor plots.

☒ A numerical value for number of cells or percentage (with statistics) is provided.

### Methodology

| | |
|---|---|
| Sample preparation | CELL THAWING/COUNTING: Quick thaw cells in 37°C water bath, Pipette cells into 5 mL PBS, rinse tube with 1 mL PBS, Spin 5 min, aspirate supernatant, Resuspend in 10 mL PBS, Count, Aliquot and spin 5 min, aspirate supernatant<br>STAINING: Zombie dye – 1:100 dilution in PBS, Resuspend cells in 66 μL of Zombie dilution, Incubate 10 min @ RT, covered, Add 5 μL Human TruStain FcX to sample, Incubate 10 min @ RT, covered; Ab cocktail: 3 μL x 8 Abs, 5 μL CCR7, Add 29 μL cocktail to sample, Incubate 15-20 min @ RT, covered<br>WASH: Add 1 mL PBS/2% FBS, Spin 5 min, aspirate supernatant, Resuspend in 1 mL PBS/2% FBS, Spin 5 min, aspirate supernatant<br>FIX: Resuspend in 500 μL FluoroFix Buffer (1% PFA), Incubate 30 min @ RT, Add 800 μL PBS/2% FBS, Spin 5 min, aspirate supernatant, Resuspend in 1 mL PBS/2% FBS, Store at 4°C until sort |
| Instrument | BD Influx and Sony MA900 |
| Software | FlowJo and Sony Cell Sorter Software Version 3.1.2 |
| Cell population abundance | Figure S1B |
| Gating strategy | Figure S1A |

☒ Tick this box to confirm that a figure exemplifying the gating strategy is provided in the Supplementary Information.

