## [Peer Review File · Nature Genetics]

Genetics and Environment Distinctively Shape the Human Immune Cell Epigenome

Corresponding Author: Professor Joseph Ecker

Version 0:

Decision Letter:

19th Feb 2025

Dear Professor Ecker,

Your Article, "Genetics and Environment Distinctively Shape the Human Immune Cell Epigenome" has now been seen by 2 referees. You will see from their comments copied below that while they find your work of considerable potential interest, they have raised quite substantial concerns that must be addressed. In light of these comments, we cannot accept the manuscript for publication, but would be interested in considering a substantially revised version that addresses these serious concerns.

We hope you will find the referees' comments useful as you decide how to proceed. If you wish to submit a substantially revised manuscript, please bear in mind that we will be reluctant to approach the referees again in the absence of major revisions.

To guide the scope of the revisions, the editors discuss the referee reports in detail within the team with a view to identifying key priorities that should be addressed in revision. In this case, we think both referees have provided constructive reviews aimed at strengthening the analyses and improving the presentation. We ask that you address the key limitations in statistical power, cohort selection, confounder adjustment, and functional validation, and address referees' technical comments as thoroughly as possible with appropriate revisions. We hope that you will find the prioritized set of referee points to be useful when revising your study.

If you choose to revise your manuscript taking into account all reviewer and editor comments, please highlight all changes in the manuscript text file. At this stage we will need you to upload a copy of the manuscript in MS Word .docx or similar editable format.

*2) If you have not done so already please begin to revise your manuscript so that it conforms to our Article format instructions, available [here](http://www.nature.com/ng/authors/article_types/index.html). Refer also to any guidelines provided in this letter.

*3) Include a revised version of any required Reporting Summary: <https://www.nature.com/documents/nr-reporting-summary.pdf>
It will be available to referees (and, potentially, statisticians) to aid in their evaluation if the manuscript goes back for peer review.

EXTENDED DATA FIGURES

Link Redacted

If you wish to submit a suitably revised manuscript we would hope to receive it within 6 months. If you cannot send it within this time, please let us know. We will be happy to consider your revision so long as nothing similar has been accepted for publication at Nature Genetics or published elsewhere. Should your manuscript be substantially delayed without notifying us in advance and your article is eventually published, the received date would be that of the revised, not the original, version.

Thank you for the opportunity to review your work.

Sincerely,
Wei

Wei Li, PhD
Senior Editor
Nature Genetics
www.nature.com/ng

Reviewers' Comments:

Reviewer #1 (Remarks to the Author):

REVIEW: Wang et al - Nature Genetics

The study by Wang et al. provides a comprehensive examination of the interplay between genetics and environment in shaping the human immune cell epigenome. A key strength of this research is its ambitious scope, utilizing single-nucleus methylation sequencing (snmC-seq2) and ATAC-seq to analyze a diverse set of exposures (viral infections like HIV-1, influenza; bacterial infections; a vaccine; and a chemical) in a significant number of individuals. This multi-omic approach allowed for a high-resolution view of epigenetic changes, revealing that environmental factors induce both hypo- and hyper-methylation (eDMRs) often at enhancer and promoter regions, while genetic variations (gDMRs) tend to influence methylation at gene bodies. The finding that these two factors target distinct genomic contexts is a novel insight, as previous studies often focused on one aspect in isolation. Furthermore previous studies did not perform such studies at single cell resolution but in bulk tissues.

Bolstering this is the additional strength is the cell-type-specific analysis, which demonstrated that the effects of both genetics and environment on methylation patterns vary significantly across different immune cell types. This granularity is crucial for understanding the nuanced mechanisms of immune regulation and disease development. Additionally, the integration of GWAS data to identify colocalizing disease-associated SNPs with meQTLs adds a layer of clinical relevance, suggesting potential pathways for how genetic risk is mediated through epigenetic changes. Even though the measurements were sparse and the sample size of individuals in this study was small, its impact as a reference atlas for such "exposome" influences on the epigenome will be widely felt.

However, the study also has certain limitations. While the range of exposures is a strength, the sample sizes for each exposure varied, and some were relatively small, which could limit the generalizability of the conclusions. The use of both

"external" healthy donors and "internal" pre-infection controls, while practical, might introduce confounders and obscure subtle differences. The study also relies on publicly available histone modification data rather than generating in-house data, which could limit the accuracy of functional inferences. Lastly no demographic metadata features in this study and would be very helpful in understanding the cohort of individuals and the components which may be attributed to ethnic differences. Could exposures to SARS-CoV-2-COVID (strain specific) and HIV in different ethnic groups produce similar DMRs or highly discordant DMRs? This atlas may provide the opportunity to ask this question and assist in clarifying any role population variation contributes to this.

Furthermore, the study is largely descriptive, and further functional assays would be needed to confirm causal links between the identified DMRs and immune response phenotypes. Despite these limitations, this research offers a highly valuable resource for understanding immune epigenomics and will likely stimulate further investigations into disease mechanisms and biomarker discovery. Several suggestions to strengthen the study below as well as some technical limitations to consider:

Major Concerns

1. Statistical Power Limitations

The study is underpowered for detecting small effect sizes, particularly in certain exposure groups. For example, the low exposure group for organophosphates (n=3) has an inadequate sample size to detect even medium effects. While moderate and large effect sizes can be detected, small effect sizes might be missed. The authors should increase sample sizes in low-exposure groups where feasible or explicitly acknowledge this limitation in the discussion. If increasing sample size is not possible, statistical techniques like meta-analysis across cohorts or Bayesian hierarchical modeling could help extract more reliable inferences from smaller groups.

2. Cohort Selection and Demographic Bias

Differences in age, sex, and genetic ancestry between exposed and control groups may confound the results. For instance, the anthrax vaccine group (likely military personnel) may differ systematically from other groups in ways that influence methylation patterns independent of exposure. The authors should provide a detailed demographic table that includes age, sex, ethnicity, and lifestyle factors for each exposure group. Additionally, using covariate adjustment techniques (such as propensity score matching or multivariable regression including demographic factors) would help mitigate these confounding effects.

3. Heterogeneity in Exposure Definition

Some exposures, particularly COVID-19 severity and OP exposure levels, are highly heterogeneous. Severe COVID cases, for example, could have experienced hospitalization, medications (e.g., steroids), and co-infections, all of which could independently influence DNA methylation beyond the viral infection itself. The authors should consider stratifying samples by exposure intensity (e.g., mild/moderate/severe COVID) and providing sensitivity analyses to assess how these covariates influence DMR findings. Adjusting for potential confounding treatments (e.g., corticosteroids) would also improve interpretability.

4. Causality vs. Association in Exposure Effects

The study interprets DMRs as exposure-induced, but some methylation differences might be pre-existing and contribute to exposure susceptibility rather than being a consequence of the exposure. The authors should include longitudinal analyses where feasible (such as using pre-exposure samples for HIV and IAV) to confirm causal relationships. Where longitudinal data is unavailable, instrumental variable approaches or Mendelian randomization could be used to strengthen causal inference.

5. Genetic Confounding in meQTL Analysis

Differences in population ancestry may confound the association between genetic variants and methylation changes (meQTLs). If ancestry is unevenly distributed across exposure groups, some observed genetic effects might be due to ancestry-related population structure rather than true genetic regulation. The authors should conduct principal component analysis (PCA) on genetic data to adjust for ancestry-related confounding in meQTL analysis. This is a standard technique in genetic association studies and would help ensure the robustness of gDMR findings.

6. Functional Validation of Regulatory Effects

The manuscript infers that exposure-associated DMRs (eDMRs) and genotype-associated DMRs (gDMRs) are functionally important, but direct experimental validation is missing. The study relies on motif enrichment analysis, which is suggestive but does not confirm actual transcription factor (TF) binding or gene expression changes. The authors should complement the analysis by incorporating expression quantitative trait locus (eQTL) analysis to test whether DMRs correlate with gene expression changes. Chromatin immunoprecipitation sequencing (ChIP-seq) for key TFs (e.g., IRF, RUNX, or CEBP families) in exposed vs. control samples would directly test whether the methylation changes affect TF binding.

7. Batch Effects in Sample Processing

The study processes different exposure groups separately, which may introduce batch effects that could drive differences in methylation rather than true biological variation. The authors should clarify the batch correction strategy used (e.g., ComBat, Harmony, or limma). If batch effects were tested and found negligible, providing a Supplementary Figure with PCA plots of methylation data across batches would help assure readers that batch artifacts were not a major concern.

8. Generalizability and External Validity

The study does not provide population ancestry details or discuss whether findings generalize across different genetic backgrounds. Given known differences in epigenetic regulation across populations, this is a potential limitation. The authors

should specify the ancestral composition of the study cohort and, if applicable, compare their gDMRs with existing multi-ethnic meQTL databases (e.g., GoDMC, BLUEPRINT). This would clarify whether the findings are likely to be broadly applicable or specific to the study population.

9. Overinterpretation of Functional Consequences

The manuscript sometimes implies that all observed methylation changes have functional consequences, but this is not directly tested. The authors should clearly distinguish between correlational findings and causal functional effects. Providing effect size estimates for DMRs would help contextualize the biological relevance of findings, and discussing potential non-functional methylation variation (e.g., passenger changes) would ensure a balanced interpretation.

Wang et al. Technical Critique of the Study

There is no doubt that this study is methodologically sophisticated, employing state-of-the-art single-cell epigenomic techniques (single-nucleus methylation sequencing snmC-seq2 and ATAC-seq) from laboratories with extended experience with these techniques. This has led to dissecting the interplay between genetics and environment in shaping immune cell epigenomes in these important contexts. Notwithstanding, there are some technical limitations and concerns that should be addressed:

1. Single-Nucleus Methylation Sequencing (snmC-seq2) – Sparse Coverage Issues

Single-cell bisulfite sequencing is inherently sparse, meaning each cell has limited CpG site coverage. CpG sites with low sequencing depth may result in erroneous differential methylation calls (false positives or false negatives). The study aggregates single-cell methylation data into pseudo-bulk profiles, which mitigates sparsity but might introduce biases (e.g. over-representing abundant cell subtypes). Are the authors able to report CpG site coverage metrics (e.g., mean coverage per site and per cell) to assure data quality. Imputation strategies (such as methylation-aware smoothing or deep-learning-based imputation methods) should be explored for refining DMR detection.

2. Cell-Type Purity and Sorting Limitations

The study relies on FACS sorting to isolate major immune cell types, but there is still intracellular heterogeneity. Some cell types, such as monocytes, show sub-clusters with different methylation profiles, which could lead to false associations between exposures and methylation if these sub-clusters differ in proportion across exposure groups. The CD14-based monocyte sorting strategy may include both classical and non-classical monocytes, which have distinct methylomes. Use single-cell clustering analysis to verify that sorted immune populations are homogeneous. Perform sub-clustering analyses within each sorted population to identify potential intra-population variability. Consider including multi-omic profiling (RNA-seq + methylation) to distinguish true cell identity.

3. ATAC-seq Integration – Lack of Matched Multi-Omics Data for All Exposures

The study only performed ATAC-seq for HIV-1 samples, meaning that the correlation between chromatin accessibility and methylation is not examined for other exposures. This raises a concern about generalizability—the reported correlation between hypo-methylation and chromatin opening may not apply to other exposures like COVID-19, MRSA, or OP exposure. Ideally, ATAC-seq should be conducted for all exposures to validate that methylation changes correspond to changes in chromatin accessibility across different conditions. If additional ATAC-seq is not feasible, the authors should acknowledge this limitation in the discussion.

4. MeQTL Mapping – Potential for False Positives

The multiple testing burden in meQTL analysis is high due to the large number of CpG sites and SNPs tested. The use of standard FDR correction (without permutation-based methods) might still lead to false positives. Many trans-meQTLs (long-range genotype-methylation associations) are reported, but these are often difficult to validate and prone to spurious associations due to population structure. The authors could use more stringent multiple testing correction (e.g., permutation-based empirical p-values). Provide validation in an independent cohort (if possible). Examine whether identified trans-meQTLs overlap with known chromatin interactions (Hi-C data) to support their validity for example with CICERO.

6. Exposure Timing and Temporal Stability of Methylation Changes

The timing of sample collection post-exposure may impact methylation findings: Acute vs. chronic infections: SARS-CoV-2 patients had different severities, but time since infection was not standardized. HIV pre/post analysis is strong, but for other exposures (e.g., OP chemicals, bacterial infections), it is unclear whether methylation changes are transient or permanent. This affects the interpretation of eDMRs—some may represent temporary immune activation states rather than long-term epigenetic memory. One workaround is to conduct follow-up sampling at multiple time points to determine whether methylation changes are stable or transient. Another is to use epigenetic clock models to determine whether observed changes align with age-related drift vs. true exposure effects. Notwithstanding these should be correctly caveated in the text.

7. Functional Relevance of DMRs – Lack of Expression Data

The study infers functional consequences from DMR enrichment at enhancers and transcription factor motifs, but does not measure gene expression changes directly. Without single-cell RNA-seq (scRNA-seq), it is unclear whether hypo- or hyper-methylation at specific sites actually alters gene expression. If possible can the authors integrate scRNA-seq or bulk RNA-seq from the same samples to determine whether DMRs correlate with transcriptional changes. Or from samples for some exposures that have undergone similar insults (e.g. Acute vs chronic) or for HIV ART type and duration. This can be bolstered by performing causal perturbation experiments (e.g., CRISPR-dCas9-DNMT/TET) to directly test whether modifying methylation at key DMRs affects gene expression.

8. Interpretation of Transcription Factor Motif Enrichment

The study infers that TF binding is disrupted based on motif enrichment at eDMRs but does not confirm actual TF occupancy. Motif enrichment does not necessarily mean that a transcription factor was actually binding to that region. There are datasets for immune cells where the TF binding using ChIP-seq has been conducted for key transcription factors (e.g., IRF, CEBP, RUNX). These can be interrogated to confirm TF occupancy.

9. Batch Effects in Methylation and ATAC-seq

The authors do not explicitly discuss batch correction methods for methylation or ATAC-seq data. Differences in library preparation, sequencing batches, or technician variability could introduce artifacts. It would be important to have an explicit statement by the authors on this. In the absence of such corrections being made they should apply appropriate batch effect correction algorithms. The authors could also show PCA plots of methylation and ATAC-seq data before and after batch correction to confirm no batch-driven clustering.

Reviewer #2 (Remarks to the Author):

A. Summary of the key results

Wang, Hariharan, Ding et al. profiled epigenome signatures of 171 PBMC samples from 110 individuals across 9 major immune cell types and uncovered pathogen and genetic factors that altered DNA methylation and chromatin accessibility.

B. Originality and significance: if not novel, please include reference

This study provides valuable insight into how host genetics and infectious agents can both influence phenotypic outcomes, as well as an incredible resource for future studies to look into context-specific meQTLs and the mechanisms by which pathogens influence disease outcomes.

C. Data & methodology: validity of approach, quality of data, quality of presentation

In general, I found the manuscript well-presented and a valuable contribution to the community. There is a lack of supplementary data supporting their reported results, for example:

- * A supplementary table listing all diseases tested and their coloc results would be helpful.
- * Summary statistics of meQTL results
- * Summary statistics of eDMRs and gDMRs

D. Appropriate use of statistics and treatment of uncertainties

- * There is a lack of description of how the authors identified DMGs. In the methods, it states it followed an online tutorial, but was it corrected for age, sex, and donor effects? Given these donors have different age, sex and genetic ancestry, I would anticipate that collectively they have a non-negligible effect on methylation levels. Could the authors clarify if they corrected for batch effects in preprocessing or modeled them as covariates during the PairwiseDMG analysis?
- * In their Method, the authors have a section on "Validation of DMRs by shuffling the samples." I could not identify the relevant information on this in the main text. Could the authors plot the null distribution generated by this approach to ensure a well-controlled false-positive rate?
- * I'm confused about the presented colocalization analysis between gDMRs and GWAS SNPs. In Fig. 6A, the authors show the distribution of meQTLs colocalized with Addison's disease SNPs. The original paper on Addison's disease GWAS [<https://doi.org/10.1038/s41467-021-21015-8>] only reported 9 genome-wide significant loci; in which case, how can the authors obtain so many colocalization signals in each cell type? Additionally, the authors performed coloc analysis in the complicated MHC region. There are some potential problems with this: (1) the known pleiotropic effect of the MHC region violates the single causal variant assumption in coloc, and (2) without including classical HLA alleles (via imputation), the authors risk omitting true causal signals. Therefore, without further validation, I question that rs910320 is the putative causal variant regulating both the meQTL and the disease.

E. Conclusions: robustness, validity, reliability

I found the term "exosome" used in the manuscript a bit too vague and would suggest the authors replace it with "infectious agents" or "pathogen" to be more precise. They may consider extending the concept to all exosomes in the discussions if they wish.

F, Suggested improvements: experiments, data for possible revision

- * Figure 1D - visually, these UMAPs of Tmem are quite "lumpy," with small islands scattered around. I wonder how robust these clusters are with different UMAP parameters (e.g., n_neighbors or min_dist). Are they distinct subpopulations (if so, that would be quite interesting) or donor effects (would be good to see them colored by donor) or other artefacts?
- * I'm curious—are monocytes 1 & 2 correlated with classical and non-classical monocytes? For example, by looking into CpG sites in CX3CR1. If so, the authors' observation regarding cMono reflects what's known in the literature.
- * It would be nice to see some QC for the SNP data, including genotype PCs, the missingness rate per SNP before imputation, the imputation r² filtering threshold, and the HWE filtering threshold. How many post-QC SNPs were available for gDMR analysis?
- * I'm aware that the donors in this study have mixed ancestry (Table S1), and the authors have corrected for this in their meQTL study. However, it would still be beneficial to confirm that none/any of the reported effects are ancestry-dependent. I might suggest running a secondary analysis of significant meQTLs in European/African samples only.

Minor comments/suggestions:

- Are there references available for the iPreX Cohort and FLU010 study (Lines 145 & 149)?
- SF1 (C) has two CD8 memory cells (Sorts 1 & 7); why?
- Please reference the previous study in Line 344.
- In the bar plot of Fig 2B, which shows different proportions of immune cells from each exposure of 7 cell types, what are the clusters at the tailoring end? For example, COVID_S seems to have some monocytes and NK cells at the end.
- Figure S6B shows an overlap between inferred SNPs but not their accuracy. The authors might consider creating a correlation plot between the frequencies of overlapping SNPs inferred by the two different methods.
- Are the number of gDMRs positively correlated with the number of cells in each cell type (Fig 5A)? It would be helpful to see a supplementary figure to illustrate this.

G. References: appropriate credit to previous work?

Yes

H. Clarity and context: lucidity of abstract/summary, appropriateness of abstract, introduction and conclusions

Clear and well-written.

Version 1:

Decision Letter:

18th Jul 2025

Dear Professor Ecker,

Your Article, "Genetics and Environment Distinctively Shape the Human Immune Cell Epigenome" has now been seen by 2 referees. You will see from their comments below that while they find your work of interest, some important points are raised by Reviewer #2. We are interested in the possibility of publishing your study in Nature Genetics, but would like to consider your response to these concerns in the form of a revised manuscript before we make a final decision on publication.

We therefore invite you to revise your manuscript taking into account all reviewer and editor comments. Please highlight all changes in the manuscript text file. At this stage we will need you to upload a copy of the manuscript in MS Word .docx or similar editable format.

*2) If you have not done so already please begin to revise your manuscript so that it conforms to our Article format instructions, available

[here](http://www.nature.com/ng/authors/article_types/index.html).

*3) Include a revised version of any required Reporting Summary: <https://www.nature.com/documents/nr-reporting-summary.pdf>

Please be aware of our [guidelines](https://www.nature.com/nature-research/editorial-policies/image-integrity) on digital image standards.

EXTENDED DATA FIGURES

Link Redacted

We hope to receive your revised manuscript within four to eight weeks. If you cannot send it within this time, please let us know.

Nature Genetics is committed to improving transparency in authorship. As part of our efforts in this direction, we are now requesting that all authors identified as 'corresponding author' on published papers create and link their Open Researcher and Contributor Identifier (ORCID) with their account on the Manuscript Tracking System (MTS), prior to acceptance. ORCID helps the scientific community achieve unambiguous attribution of all scholarly contributions. You can create and link your ORCID from the home page of the MTS by clicking on 'Modify my Springer Nature account'. For more information please visit please visit www.springernature.com/orcid.

Sincerely,
Wei

Wei Li, PhD
Senior Editor
Nature Genetics
www.nature.com/ng

Reviewers' Comments:

Reviewer #1 (Remarks to the Author):

The authors have responded to the majority of my remarks satisfactorily.

Reviewer #2 (Remarks to the Author):

I appreciate the authors' efforts in addressing my previous comments, particularly where substantial revisions to the analyses and reporting have been made. The manuscript has improved as a result. I have two remaining areas of concern: **1. Colocalization Analysis.** While the authors have expanded on their colocalization methodology, several points remain unclear or require further clarification:

- a. Missing chromosome names (chr1?) in Figure 6B
- b. Lack of references: The GWAS and eQTL summary statistics used in Figure 6B are not cited.
- c. Threshold for colocalization ($PP4 > 0.5$): The use of $PP4 > 0.5$ as a threshold to claim colocalization is questionable. While a low $PP4$ indicates a lack of shared causality, a $PP4 > 0.5$ does not necessarily imply high confidence in colocalization. The accepted standard in the field is generally $PP4 > 0.7/0.8$ to indicate strong evidence. I recommend revisiting this threshold and providing justification if deviating from commonly accepted practices.
- d. Details of Figure S9B: More information is needed regarding the heatmap shown in Figure S9B. Specifically: (a) What input data or results were used to construct the heatmap? (b) Why is Eczema shown as the most enriched phenotype? This is unexpected (?) and would benefit from additional explanation or discussion in the main text.
- e. Gene prioritization at the locus (EFEMP2 vs. OVOL1): In the main text, EFEMP2 is proposed as the potentially causal gene. However, OVOL1 is the nearest gene and appears to be regulated by the lead variant based on OpenTargets (https://platform.opentargets.org/variant/11_65791795_A_G). It is unclear why EFEMP2 is prioritized over OVOL1 - maybe additional colocalization or multi-trait colocalization (e.g., moloc) analysis could help strengthen the case for the prioritized gene.

2. Genetic PCs. In Figure R9, the principal components are labeled as PC2 vs. PC3. Is this a mislabeling? What happened to PC1?

Additionally, I want to raise a point regarding the terminology used to describe genetic ancestry. The manuscript currently refers to donors as "Black" and "White" based on their inferred genetic ancestry. As the field increasingly distinguishes between race, ethnicity and genetic ancestry, I strongly recommend that the authors adopt more precise and appropriate terminology, e.g. "individuals of African genetic ancestry" and "individuals of European genetic ancestry."

Version 2:

Decision Letter:

15th Aug 2025

Dear Professor Ecker,

Your Article, "Genetics and Environment Distinctively Shape the Human Immune Cell Epigenome" has now been seen by 1 referees. You will see from their comments below that while they find your work of interest, some important points are raised. We are interested in the possibility of publishing your study in Nature Genetics, but would like to consider your response to these concerns in the form of a revised manuscript before we make a final decision on publication.

We therefore invite you to revise your manuscript taking into account all reviewer and editor comments. Please highlight all changes in the manuscript text file. At this stage we will need you to upload a copy of the manuscript in MS Word .docx or similar editable format.

*2) If you have not done so already please begin to revise your manuscript so that it conforms to our Article format instructions, available

[here](http://www.nature.com/ng/authors/article_types/index.html).

*3) Include a revised version of any required Reporting Summary: <https://www.nature.com/documents/nr-reporting-summary.pdf>

Please be aware of our [guidelines](https://www.nature.com/nature-research/editorial-policies/image-integrity) on digital image standards.

EXTENDED DATA FIGURES

Link Redacted

We hope to receive your revised manuscript within four to eight weeks. If you cannot send it within this time, please let us know.

Sincerely,
Wei

Wei Li, PhD
Senior Editor
Nature Genetics
www.nature.com/ng

Reviewers' Comments:

Reviewer #2 (Remarks to the Author):

I thank the authors for their detailed responses. I have one final question regarding the genetic PCs. Something appears to be off here: typically, the first genetic PC captures major continental population structure. In the presented figure, PC1 does not appear to be centred around zero, which may suggest that the genotypes were not centred prior to PCA. Could the authors please recheck their preprocessing steps to confirm PCA was performed appropriately?

Version 3:

Decision Letter:

Our ref: NG-A67664R2

19th Sep 2025

Dear Dr. Ecker,

Thank you for submitting your revised manuscript "Genetics and Environment Distinctively Shape the Human Immune Cell Epigenome" (NG-A67664R2). It has now been seen by the original referees and their comments are below. The reviewers find that the paper has improved in revision, and therefore we'll be happy in principle to publish it in Nature Genetics, pending minor revisions to comply with our editorial and formatting guidelines.

Sincerely,
Wei

Wei Li, PhD
Senior Editor
Nature Genetics
www.nature.com/ng

Reviewer #2 (Remarks to the Author):

I'm glad to see that the authors have corrected their PC analysis, and this change does not change the main conclusions of the manuscript.

I have no further comments.

Reviewer #1 (Remarks to the Author):

REVIEW: Wang et al - Nature Genetics

The study by Wang et al. provides a comprehensive examination of the interplay between genetics and environment in shaping the human immune cell epigenome. A key strength of this research is its ambitious scope, utilizing single-nucleus methylation sequencing (snmC-seq2) and ATAC-seq to analyze a diverse set of exposures (viral infections like HIV-1, influenza; bacterial infections; a vaccine; and a chemical) in a significant number of individuals. This multi-omic approach allowed for a high-resolution view of epigenetic changes, revealing that environmental factors induce both hypo- and hyper-methylation (eDMRs) often at enhancer and promoter regions, while genetic variations (gDMRs) tend to influence methylation at gene bodies. The finding that these two factors target distinct genomic contexts is a novel insight, as previous studies often focused on one aspect in isolation. Furthermore previous studies did not perform such studies at single cell resolution but in bulk tissues.

Bolstering this is the additional strength is the cell-type-specific analysis, which demonstrated that the effects of both genetics and environment on methylation patterns vary significantly across different immune cell types. This granularity is crucial for understanding the nuanced mechanisms of immune regulation and disease development. Additionally, the integration of GWAS data to identify colocalizing disease-associated SNPs with meQTLs adds a layer of clinical relevance, suggesting potential pathways for how genetic risk is mediated through epigenetic changes. Even though the measurements were sparse and the sample size of individuals in this study was small, its impact as a reference atlas for such “exposome” influences on the epigenome will be widely felt.

Reply:

Thank you for your insightful feedback! We appreciate your recognition of the importance of our multi-omic approach and single-cell resolution in differentiating the impact of genetic and environmental factors on the immune cell epigenome. Your observation regarding the distinct genomic targeting by eDMRs and gDMRs underscores a key finding of our research. Although we acknowledge the limitations posed by sparse measurements and sample size, we believe this work will serve as a valuable resource for future investigations into the exposome and epigenome. We are grateful for your feedback!

However, the study also has certain limitations. While the range of exposures is a strength, the sample sizes for each exposure varied, and some were relatively small,

which could limit the generalizability of the conclusions. The use of both "external" healthy donors and "internal" pre-infection controls, while practical, might introduce confounders and obscure subtle differences. The study also relies on publicly available histone modification data rather than generating in-house data, which could limit the accuracy of functional inferences. Lastly no demographic metadata features in this study and would be very helpful in understanding the cohort of individuals and the components which may be attributed to ethnic differences. Could exposures to SARS-CoV2-COVID (strain specific) and HIV in different ethnic groups produce similar DMRs or highly discordant DMRs? This atlas may provide the opportunity to ask this question and assist in clarifying any role population variation contributes to this. Furthermore, the study is largely descriptive, and further functional assays would be needed to confirm causal links between the identified DMRs and immune response phenotypes. Despite these limitations, this research offers a highly valuable resource for understanding immune epigenomics and will likely stimulate further investigations into disease mechanisms and biomarker discovery. Several suggestions to strengthen the study below as well as some technical limitations to consider:

Reply:

Thank you for your insightful feedback! We appreciate your concerns regarding sample size variation, potential confounders, and the importance of incorporating demographic metadata.

We appreciate the reviewer's thoughtful comment regarding the use of both external healthy donors and internal pre-infection controls. We acknowledge that this design, while practical and necessary given sample availability, may introduce heterogeneity that could obscure subtle, exposure-specific differences. Specifically, baseline inter-individual variability between external donors and internal controls may confound the interpretation of mild exposure-induced changes. While we aimed to mitigate this by careful batch correction and statistical modeling, we recognize that this limitation may affect the sensitivity for detecting subtle effects and should be considered when interpreting the findings.

According to your insightful suggestion, we performed eDMR analysis across different ethnic groups and discovered significant differences between Black and White individuals, highlighting the importance of considering population-specific epigenetic responses to environmental exposures.

Although our study is primarily descriptive, we recognize the importance of functional validation to establish causal relationships. Ideally, we would generate matched histone ChIP-seq data for these samples; however, due to sample limitations, we instead relied on high-quality ENCODE datasets. We acknowledge and apologize for any prior over-

interpretation of our findings. In the revised manuscript, we have clearly stated that our results are associative in nature and do not imply causality. We hope that this atlas will nonetheless serve as a valuable resource for advancing immune epigenomics research across diverse populations.

Major Concerns

1. Statistical Power Limitations

The study is underpowered for detecting small effect sizes, particularly in certain exposure groups. For example, the low exposure group for organophosphates (n=3) has an inadequate sample size to detect even medium effects. While moderate and large effect sizes can be detected, small effect sizes might be missed. The authors should increase sample sizes in low-exposure groups where feasible or explicitly acknowledge this limitation in the discussion. If increasing sample size is not possible, statistical techniques like meta-analysis across cohorts or Bayesian hierarchical modeling could help extract more reliable inferences from smaller groups.

Reply:

We thank the reviewer for raising this concern. We acknowledge that the sample size for the low-exposure organophosphate group (n=3) is indeed small and limits our power to detect subtle effects of low-level exposure. In the revised manuscript, we have explicitly acknowledged this limitation in the Discussion section. We also emphasize that our study is powered to detect moderate to large effect sizes, which are more likely to be biologically relevant.

2. Cohort Selection and Demographic Bias

Differences in age, sex, and genetic ancestry between exposed and control groups may confound the results. For instance, the anthrax vaccine group (likely military personnel) may differ systematically from other groups in ways that influence methylation patterns independent of exposure. The authors should provide a detailed demographic table that includes age, sex, ethnicity, and lifestyle factors for each exposure group. Additionally, using covariate adjustment techniques (such as propensity score matching or multivariable regression including demographic factors) would help mitigate these confounding effects.

Reply:

We thank the reviewer for this important suggestion. In response, we have now included a detailed demographic table (age, sex, and ethnicity) for each exposure group in the revised manuscript [Table S1]. For the donors we didn't have Ethnicity information, we inferred this from the genotype PCA analysis. Additionally, we used demographic variables—including age, sex, and ethnicity—as covariates in our statistical models

when identifying exposure-associated differentially methylated regions (eDMRs), in order to account for potential confounding effects. While we acknowledge that certain groups (e.g., the anthrax vaccine group) may still differ systematically in unmeasured ways, our approach helps to mitigate known sources of bias and improve the robustness of our findings.

3. Heterogeneity in Exposure Definition

Some exposures, particularly COVID-19 severity and OP exposure levels, are highly heterogeneous. Severe COVID cases, for example, could have experienced hospitalization, medications (e.g., steroids), and co-infections, all of which could independently influence DNA methylation beyond the viral infection itself. The authors should consider stratifying samples by exposure intensity (e.g., mild/moderate/severe COVID) and providing sensitivity analyses to assess how these covariates influence DMR findings. Adjusting for potential confounding treatments (e.g., corticosteroids) would also improve interpretability.

Reply:

We acknowledge the reviewer's insightful comment regarding the heterogeneity of COVID-19 severity and its potential confounding effects on DNA methylation. In our study, we stratified COVID-19 samples into severe and non-severe groups based on clinical criteria to address this concern. However, due to the lack of available data on hospitalization and medication use, we were unable to perform sensitivity analyses to assess the influence of these covariates on our DMR findings. While we acknowledge that residual confounding may still exist, we believe our approach adequately addresses this issue and improves the robustness of our conclusions, we have discussed this limitation in the revised manuscript.

4. Causality vs. Association in Exposure Effects

The study interprets DMRs as exposure-induced, but some methylation differences might be pre-existing and contribute to exposure susceptibility rather than being a consequence of the exposure. The authors should include longitudinal analyses where feasible (such as using pre-exposure samples for HIV and IAV) to confirm causal relationships. Where longitudinal data is unavailable, instrumental variable approaches or Mendelian randomization could be used to strengthen causal inference.

Reply:

We thank the reviewer for this insightful comment. We agree that distinguishing between pre-existing methylation patterns that may predispose individuals to infection versus those that result from exposure is important for causal interpretation. However, our primary focus was on identifying differentially methylated regions (eDMRs) that are associated with exposure status,

not necessarily induced by the exposure. We have revised our manuscript to minimize the risk of overinterpretation.

Where feasible, we included longitudinal comparisons (e.g., pre- and post-infection samples for HIV and IAV) to provide additional support for exposure-associated changes. However, such data were not uniformly available across all exposure types. While we appreciate the suggestion to use Mendelian Randomization (MR), we note that MR is not well-suited for this context, as infections like HIV and IAV are not genetically determined exposures, and valid genetic instruments are lacking. Therefore, causal inference through MR is not feasible in this setting. But in our externally controlled exposures, those eDMRs are only different between exposed donors and controls, but not different between different sets of controls.

We have clarified these points in the Discussion and added this as a limitation of the study.

5. Genetic Confounding in meQTL Analysis

Differences in population ancestry may confound the association between genetic variants and methylation changes (meQTLs). If ancestry is unevenly distributed across exposure groups, some observed genetic effects might be due to ancestry-related population structure rather than true genetic regulation. The authors should conduct principal component analysis (PCA) on genetic data to adjust for ancestry-related confounding in meQTL analysis. This is a standard technique in genetic association studies and would help ensure the robustness of gDMR findings.

Reply:

We thank the reviewer for raising this important point. To account for potential confounding due to population stratification, we performed principal component analysis (PCA) on the genotype data and included the top five genetic principal components as covariates in all meQTL analyses. This standard correction helps mitigate ancestry-related confounding and supports the robustness of our gDMR findings. We have now clarified this step in the Methods section.

6. Functional Validation of Regulatory Effects

The manuscript infers that exposure-associated DMRs (eDMRs) and genotype-associated DMRs (gDMRs) are functionally important, but direct experimental validation is missing. The study relies on motif enrichment analysis, which is suggestive but does not confirm actual transcription factor (TF) binding or gene expression changes. The authors should complement the analysis by incorporating expression quantitative trait locus (eQTL) analysis to test whether DMRs correlate with gene expression changes. Chromatin immunoprecipitation sequencing (ChIP-seq) for key TFs (e.g., IRF, RUNX, or

CEBP families) in exposed vs. control samples would directly test whether the methylation changes affect TF binding.

Reply:

We thank the reviewer for the insightful suggestions regarding the functional exploration of eDMRs and gDMRs.

To investigate potential regulatory mechanisms, we performed Summary-based Mendelian Randomization (SMR) analysis integrating our meQTL and with whole blood eQTL data from the eQTLGen Consortium. By setting the exposure as gDMR and outcome as gene expression in the SMR analysis, we tested whether genetic variants associated with gDMRs are also associated with expression levels of nearby genes, providing evidence for potential causal relationships. We did not conduct HEIDI analysis, so this does not rule out the possibility that some of the DMR-gene associations identified in our SMR analysis are due to linkage rather than pleiotropy or causality. Due to the limited sample size of the white only cohort, the p-value threshold required to test a probe (DMR) for an association was lowered to 5×10^{-6} instead of the default 5×10^{-8} . This analysis revealed that around 30% of the tested DMRs were identified by SMR to have a potential causal relationship with a gene in the mixed population meQTL while around ~20% of the tested DMRs were identified to have a potential causal relationship in the white population meQTL, supporting the possibility that methylation changes at certain loci may mediate gene expression (Figure R1).

Figure R1. ratios of the gDMRs in each cell type that are associated with gene expression. Left panel: meQTLs from all donors; right: meQTLs from White population only.

While we are currently unable to perform ChIP-seq experiments for the transcription factors (TFs) of interest, we leveraged publicly available ChIP-seq datasets to assess the overlap between our eDMRs and TF binding sites. We conducted enrichment analysis using these datasets and observed significant enrichment of our eDMRs within ChIP-seq peaks for TFs whose motifs were also enriched in our data (Figure S6D), providing orthogonal support for their potential functional relevance.

Figure S6D. enrichment of eDMRs in TF ChIP-seq peaks in corresponding cell types.

7. Batch Effects in Sample Processing

The study processes different exposure groups separately, which may introduce batch effects that could drive differences in methylation rather than true biological variation. The authors should clarify the batch correction strategy used (e.g., ComBat, Harmony, or limma). If batch effects were tested and found negligible, providing a Supplementary Figure with PCA plots of methylation data across batches would help assure readers that batch artifacts were not a major concern.

Reply:

We appreciate your concern regarding potential batch effects. We have carefully addressed this issue by applying Harmony, a widely-used batch correction algorithm, to our methylation data. Following batch correction, we performed principal component analysis (PCA) and confirmed that batch effects were negligible and did not drive clustering of samples (Figure S3A). We have now included an explicit statement in the Methods section describing our batch correction strategy and referenced the Supplementary Figure for visual confirmation.

Figure S3A. T-SNE plot of the clustering in each cell type after harmony integration by donor.

8. Generalizability and External Validity

The study does not provide population ancestry details or discuss whether findings generalize across different genetic backgrounds. Given known differences in epigenetic regulation across populations, this is a potential limitation. The authors should specify the ancestral composition of the study cohort and, if applicable, compare their gDMRs with existing multi-ethnic meQTL databases (e.g., GoDMC, BLUEPRINT). This would clarify whether the findings are likely to be broadly applicable or specific to the study population.

Reply:

Thank you for raising the important point regarding ancestral composition. We have carefully considered this and performed additional analyses to address your concern about the generalizability of our meQTL findings.

Our dataset includes individuals of White, Black, Hispanic. To demonstrate the robustness of our meQTLs across different genetic backgrounds, we conducted a separate meQTL analysis specifically within the White population.

The results of this analysis show a highly significant overlap between the DMRs identified in the White-only meQTL analysis and those identified in the analysis of the mixed ancestry population (Supplementary Table R2, hypergeometric p-value < 1e-26). This strong overlap provides compelling evidence that our findings are broadly generalizable across the diverse genetic backgrounds present in our cohort.

Regarding the comparison with public meQTL databases (GoDMC and EPIC_DB), we observed limited overlap. We believe this discrepancy can be attributed to several factors:

1. **Genetic and Environmental Differences:** The GoDMC and EPIC_DB cohorts primarily consist of individuals of European and British ancestry, which may differ both genetically and environmentally from the White American population included in our study.
2. **Methodological Variations:** These databases utilized the EPIC and HM450 DNA methylation arrays, which assay a limited number of CpG sites in bulk tissue. In contrast, our study employed whole-genome bisulfite sequencing in single cells, providing comprehensive methylation data across the genome. Furthermore, our analysis focused on differentially methylated regions (DMRs), while the public databases often report meQTLs at the single-CpG level.

We believe these differences in ancestral composition and methodology likely explain the limited overlap observed and further highlight the novel insights provided by our whole-genome sequencing approach in a more diverse population.

cell_type	#DMR	#DMR in White meQTL	#DMR in All meQTL	#overlapped DMR between White and All	hypergeometric p-value
B-Mem	701087	13808	163024	8341	0
Tc-Naive	1468741	21669	237220	12819	0
B-Naive	851460	12771	155230	7597	0
Th-Naive	1473817	21794	241838	13215	0
Monocyte	1304246	16094	256093	9418	0
Th-Mem	1306355	19955	249337	12492	0
Tc-Mem	2929519	52746	742689	30732	0
NK-cell1	435140	12171	150185	7346	0
NK-cell2	898401	12982	171116	7725	0

Figure R2. hypergeometric test on the two sets of meQTLs based on all donors and white population only.

9. Overinterpretation of Functional Consequences

The manuscript sometimes implies that all observed methylation changes have functional consequences, but this is not directly tested. The authors should clearly distinguish between correlational findings and causal functional effects. Providing effect size estimates for DMRs would help contextualize the biological relevance of findings, and discussing potential non-functional methylation variation (e.g., passenger changes) would ensure a balanced interpretation.

Reply:

We appreciate the reviewer's thoughtful comment. We agree that methylation changes identified in this study are correlational and do not, on their own, establish functional or causal effects. We have revised the manuscript to more clearly distinguish between association and causation, particularly in the interpretation of eDMRs. To help contextualize the biological relevance of our findings, we have calculated and now report effect size estimates for the eDMRs. Additionally, we have added discussion acknowledging that some methylation changes may be non-functional or reflect passenger events, particularly in highly dynamic or heterogeneous cellular contexts. These clarifications aim to provide a more balanced interpretation of the results.

Wang et al. Technical Critique of the Study

There is no doubt that this study is methodologically sophisticated, employing state-of-the-art single-cell epigenomic techniques (single-nucleus methylation sequencing snmC-seq2 and ATAC-seq) from laboratories with extended experience with these techniques. This has led to dissecting the interplay between genetics and environment in shaping immune cell epigenomes in these important contexts. Notwithstanding, there are some technical limitations and concerns that should be addressed:

1. Single-Nucleus Methylation Sequencing (snmC-seq2) – Sparse Coverage Issues

Single-cell bisulfite sequencing is inherently sparse, meaning each cell has limited CpG site coverage. CpG sites with low sequencing depth may result in erroneous differential methylation calls (false positives or false negatives). The study aggregates single-cell methylation data into pseudo-bulk profiles, which mitigates sparsity but might introduce biases (e.g. over-representing abundant cell subtypes). Are the authors able to report CpG site coverage metrics (e.g., mean coverage per site and per cell) to assure data quality. Imputation strategies (such as methylation-aware smoothing or deep-learning-based imputation methods) should be explored for refining DMR detection.

Reply:

We have added a supplementary figure (Figure S2) to show quality control metrics at the cell, sample, and condition levels to address concerns about coverage. Additionally, we acknowledge the issue of sparse coverage in single-cell bisulfite sequencing, which may lead to erroneous differential methylation calls. While aggregating single-cell methylation data into pseudo-bulk profiles helps mitigate sparsity, it may introduce biases such as over-representing abundant cell subtypes. The number of cells from each donor in each exposure is balanced in our sorting strategy, ensuring representation across conditions and minimizing potential biases.

We attempted using DSS, a smoothing method, to call DMRs; however, the DMRs identified by DSS (Figure R3) were not as convincing as those obtained using our current method (Figure R4).

Figure S2. Quality control of the single-nucleus methylation sequencing data and merged pseudo bulk for each sample and condition

Figure R3. Column normalized methylation levels DMRs between HIV-1 samples using DSS smoothing

Figure R4. Column normalized methylation levels at DMRs using our current strategy

2. Cell-Type Purity and Sorting Limitations

The study relies on FACS sorting to isolate major immune cell types, but there is still intracellular heterogeneity. Some cell types, such as monocytes, show sub-clusters with different methylation profiles, which could lead to false associations between exposures and methylation if these sub-clusters differ in proportion across exposure groups. The CD14-based monocyte sorting strategy may include both classical and non-classical monocytes, which have distinct methylomes. Use single-cell clustering analysis to verify that sorted immune populations are homogeneous. Perform sub-clustering analyses within each sorted population to identify potential intra-population variability. Consider including multi-omic profiling (RNA-seq + methylation) to distinguish true cell identity.

Reply:

We thank the reviewer for raising this important point. As noted, we performed within-cell-type clustering for each FACS-sorted population to assess potential intra-population heterogeneity. These analyses indeed revealed substructure in some cell types, such as monocytes. To address the concern that differential subcluster composition might confound methylation-exposure associations, we examined the relative proportions of identified subclusters across exposure groups (Figure 2). While some variability was observed, these differences did not consistently track with exposure status and did not account for the major methylation changes reported.

Regarding cell identity, our sorting strategy relied on canonical surface markers, and in the case of CD14+ monocytes, they are not likely classical and non-classical monocytes based on the methylation levels at marker genes (Figure 2G).

Figure 2G. Methylation levels at monocyte markers in the two clusters of monocytes.

3. ATAC-seq Integration – Lack of Matched Multi-Omics Data for All Exposures

The study only performed ATAC-seq for HIV-1 samples, meaning that the correlation between chromatin accessibility and methylation is not examined for other exposures. This raises a concern about generalizability—the reported correlation between hypomethylation and chromatin opening may not apply to other exposures like COVID-19, MRSA, or OP exposure. Ideally, ATAC-seq should be conducted for all exposures to

validate that methylation changes correspond to changes in chromatin accessibility across different conditions. If additional ATAC-seq is not feasible, the authors should acknowledge this limitation in the discussion.

Reply:

We thank the reviewer for pointing out this important issue. While our initial submission included ATAC-seq data only for HIV-1–exposed samples, we have now expanded the dataset to include ATAC-seq profiles for all other exposures analyzed in this study (Figure S7D-G), with the exception of MRSA. This addition allows us to systematically examine the relationship between chromatin accessibility and methylation across a broader range of environmental conditions.

Figure S7D-E. The overlap between eDMR and DAR in each exposure. D. Anthrax vaccine. E. OP. F. SARS-Cov2. G. IAV.

Unlike our earlier findings in HIV-1, where we observed significant overlap between hypomethylation and increased chromatin accessibility, we see less overlap across other exposures, including COVID-19 and OP. This discrepancy is likely due to the fact that, in the HIV-1 cohort methylation and ATAC sequencing, the same donors were sampled both before and after exposure. However, we acknowledge that without ATAC-seq data for MRSA-exposed samples, we cannot formally assess this relationship in that context. We have added a note in the Discussion section to highlight this limitation and emphasize the importance of future profiling of MRSA-exposed samples to complete the cross-exposure comparison.

4. MeQTL Mapping – Potential for False Positives

The multiple testing burden in meQTL analysis is high due to the large number of CpG sites and SNPs tested. The use of standard FDR correction (without permutation-based

methods) might still lead to false positives. Many trans-meQTLs (long-range genotype-methylation associations) are reported, but these are often difficult to validate and prone to spurious associations due to population structure. The authors could use more stringent multiple testing correction (e.g., permutation-based empirical p-values). Provide validation in an independent cohort (if possible). Examine whether identified trans-meQTLs overlap with known chromatin interactions (Hi-C data) to support their validity for example with CICERO.

Reply:

We thank the reviewer for their valuable suggestions regarding the control of false positives in trans-meQTL analysis. We agree that trans associations, due to the vast number of variant-CpG site pairs tested, require careful multiple testing correction.

In our study, we performed trans-meQTL mapping using TensorQTL in trans mode, which computes nominal p-values using linear regression. To control for multiple testing, we applied a stringent Bonferroni-style correction, setting the significance threshold as $FDR < 0.05$. This approach is highly conservative and helps reduce the likelihood of false positives even in the absence of permutation-based empirical p-values. While permutation testing offers a more nuanced estimation of the null distribution, it is computationally intensive and not directly supported in TensorQTL's trans mode. As such, implementing full-scale permutations was beyond the scope of the current study.

We present the trans-meQTL results as statistically stringent, hypothesis-generating findings to guide future work, which may include validation in independent cohorts or integration with chromatin interaction data.

We also examined whether our trans-meQTL-gDMR pairs overlapped with chromatin loops identified from Hi-C data from ENCODE. However, we found very limited overlap. This suggests that the SNPs associated with distal gDMRs may not necessarily engage in direct physical contact with these regions, at least as captured by current chromatin conformation datasets. It is possible that regulatory interactions underlying trans-meQTLs occur via indirect mechanisms or through chromatin dynamics that are not fully resolved by bulk Hi-C data.

6. Exposure Timing and Temporal Stability of Methylation Changes

The timing of sample collection post-exposure may impact methylation findings: Acute vs. chronic infections: SARS-CoV-2 patients had different severities, but time since infection was not standardized. HIV pre/post analysis is strong, but for other exposures (e.g., OP chemicals, bacterial infections), it is unclear whether methylation changes are transient or permanent. This affects the interpretation of eDMRs—some may represent temporary immune activation states rather than long-term epigenetic memory. One workaround is to conduct follow-up sampling at multiple time points to determine

whether methylation changes are stable or transient. Another is to use epigenetic clock models to determine whether observed changes align with age-related drift vs. true exposure effects. Notwithstanding these should be correctly caveated in the text.

Reply:

We thank the reviewer for this important point. While HIV samples included pre- and post-exposure comparisons, standardized timing was not available for other exposures such as SARS-CoV-2, OP chemicals, or bacterial infections. As such, we cannot fully distinguish transient immune responses from lasting epigenetic changes, and we have noted this limitation in the Discussion.

To evaluate whether eDMRs reflect age-related epigenetic drift, we examined their proximity to age-associated DMRs within our dataset. We found no significant enrichment compared to randomly selected CpG sites. However, in certain cell types, a greater number of eDMRs did colocalize with age-associated DMRs. These findings suggest that, overall, most eDMRs are not driven by age-related methylation changes. Future longitudinal studies will be essential to determine the stability and persistence of these epigenomic alterations over time.

Figure R5. Distance of age-related and random CpG sites to the eDMRs

7. Functional Relevance of DMRs – Lack of Expression Data

The study infers functional consequences from DMR enrichment at enhancers and transcription factor motifs, but does not measure gene expression changes directly. Without single-cell RNA-seq (scRNA-seq), it is unclear whether hypo- or hyper-methylation at specific sites actually alters gene expression. If possible can the authors integrate scRNA-seq or bulk RNA-seq from the same samples to determine whether

DMRs correlate with transcriptional changes. Or from samples for some exposures that have undergone similar insults (e.g. Acute vs chronic) or for HIV ART type and duration. This can be bolstered by performing causal perturbation experiments (e.g., CRISPR-dCas9-DNMT/TET) to directly test whether modifying methylation at key DMRs affects gene expression.

Reply:

We agree that integrating gene expression data would strengthen the interpretation of DMRs. While we did not have matched RNA-seq data from the same samples, we leveraged publicly available datasets from HIV and COVID-19 studies to assess functional relevance. For HIV-1 single-cell RNA-seq data, we use the differentially expressed genes (DEGs) from Kazer et al. (PMID:32251406). For COVID-19, we downloaded the single-cell RNA-seq data from Ren et al. (PMID: 33657410), and performed DEG analysis with scanpy.

We found that eDMRs were significantly more likely to be located near differentially expressed genes (DEGs) compared to both random genes and random CpG sites in T cells and monocytes. While eDMRs from NK cells are less likely to be related with DEGs. These results support a link between methylation changes and transcriptional regulation

Figure R6. Wilcoxon rank-sum test (Mann–Whitney U test) on the number of eDMRs within 1 Mb to the DEGs and random genes (R6A, C), or number of eDMRs to DEGs and number of random CpG sites to DEGs (R6B, D).

We have added these findings to the manuscript and now highlight in the Discussion that future studies with paired methylation and RNA-seq, as well as perturbation experiments, will be important to establish causality.

8. Interpretation of Transcription Factor Motif Enrichment

The study infers that TF binding is disrupted based on motif enrichment at eDMRs but does not confirm actual TF occupancy. Motif enrichment does not necessarily mean that a transcription factor was actually binding to that region. There are datasets for immune cells where the TF binding using ChIP-seq has been conducted for key transcription factors (e.g., IRF, CEBP, RUNX). These can be interrogated to confirm TF occupancy.

Reply:

We agree that motif enrichment alone does not confirm transcription factor binding. To address this, we examined publicly available ChIP-seq datasets in immune cells and found that eDMRs with enriched motifs (e.g., IRF, CEBP, RUNX) also show significant overlap with corresponding TF binding peaks (Figure S6D). This supports the idea that the methylation changes occur at functionally relevant regulatory sites. We have added this analysis to the manuscript and clarified it in the Results and Discussion sections.

9. Batch Effects in Methylation and ATAC-seq

The authors do not explicitly discuss batch correction methods for methylation or ATAC-seq data. Differences in library preparation, sequencing batches, or technician variability could introduce artifacts. It would be important to have an explicit statement by the authors on this. In the absence of such corrections being made they should apply appropriate batch effect correction algorithms. The authors could also show PCA plots of methylation and ATAC-seq data before and after batch correction to confirm no batch-driven clustering.

Reply:

We thank the reviewer for raising this important point. To account for potential batch effects, we applied Harmony during integration of both methylation and ATAC-seq data. As shown in the newly added UMAP plots (Figure S7A-B), batch-driven clustering was evident before correction and substantially reduced after applying Harmony. We now explicitly describe this step in the Methods and highlight it in the Results.

Figure S7A-B. UMAP of ATAC and methylation data before (A) and after (B) Harmony integration.

Reviewer #2 (Remarks to the Author):

A. Summary of the key results

Wang, Hariharan, Ding et al. profiled epigenome signatures of 171 PBMC samples from 110 individuals across 9 major immune cell types and uncovered pathogen and genetic factors that altered DNA methylation and chromatin accessibility.

B. Originality and significance: if not novel, please include reference

This study provides valuable insight into how host genetics and infectious agents can both influence phenotypic outcomes, as well as an incredible resource for future studies to look into context-specific meQTLs and the mechanisms by which pathogens influence disease outcomes.

Reply:

We appreciate the reviewer's positive feedback. While previous studies have separately explored genetic and environmental influences, our work uniquely integrates these factors with multi-omic epigenetic profiling, providing a valuable resource for future studies on context-specific meQTLs and pathogen-driven mechanisms in disease.

C. Data & methodology: validity of approach, quality of data, quality of presentation

In general, I found the manuscript well-presented and a valuable contribution to the community. There is a lack of supplementary data supporting their reported results, for example:

- * A supplementary table listing all diseases tested and their coloc results would be helpful.
- * Summary statistics of meQTL results
- * Summary statistics of eDMRs and gDMRs

Reply:

We thank the reviewer for their positive feedback. We agree that providing supplementary data would enhance the transparency and reproducibility of our results. We have now included the following supplementary materials:

- A table listing all diseases tested and their coloc results
- Summary statistics for meQTL results
- Summary statistics for eDMRs and gDMRs

These additions are intended to provide further clarity and support the reported findings

D. Appropriate use of statistics and treatment of uncertainties

* There is a lack of description of how the authors identified DMGs. In the methods, it states it followed an online tutorial, but was it corrected for age, sex, and donor effects? Given these donors have different age, sex and genetic ancestry, I would anticipate that collectively they have a non-negligible effect on methylation levels. Could the authors clarify if they corrected for batch effects in preprocessing or modeled them as covariates during the PairwiseDMG analysis?

Reply:

We thank the reviewer for this valuable suggestion. In our original analysis, we did not include age, sex, and ethnicity as covariates. In response to this comment, we have now performed a linear regression analysis incorporating these covariates to identify genes associated with the two monocyte clusters. The updated results revealed substantial overlap between the differentially methylated genes (DMGs) and the cluster-associated genes (Figure R7). Accordingly, we have replaced the original DMG results with those derived from this covariate-adjusted analysis in the revised manuscript.

However, for the analysis of DMGs between monocyte clusters in the HIV-1 exposure condition, all donors were male, aged 20–30, and of Hispanic background. Given this demographic homogeneity, we did not repeat the linear regression for that specific comparison.

DMGs between two clusters of Monocytes

Figure R7. Venn diagram shows the overlap of DMGs using pairwise comparison and linear regression using age, sex and ethnicity as covariates.

* In their Method, the authors have a section on “Validation of DMRs by shuffling the samples.” I could not identify the relevant information on this in the main text. Could the authors plot the null distribution generated by this approach to ensure a well-controlled false-positive rate?

Reply:

We apologize for missing this in the main text. We have now added this in the revised manuscript, and the null distribution was shown in Figure S5D.

Figure S5D. Distribution of HIV-1 eDMR p values and DMRs from shuffled labels.

* I'm confused about the presented colocalization analysis between gDMRs and GWAS SNPs. In Fig. 6A, the authors show the distribution of meQTLs colocalized with Addison's disease SNPs. The original paper on Addison's disease GWAS [<https://doi.org/10.1038/s41467-021-21015-8>] only reported 9 genome-wide significant loci; in which case, how can the authors obtain so many colocalization signals in each cell type? Additionally, the authors performed coloc analysis in the complicated MHC region. There are some potential problems with this: (1) the known pleiotropic effect of the MHC region violates the single causal variant assumption in coloc, and (2) without including classical HLA alleles (via imputation), the authors risk omitting true causal signals. Therefore, without further validation, I question that rs910320 is the putative causal variant regulating both the meQTL and the disease.

Reply:

Thank you for pointing out the error regarding the colocalization analysis. We have addressed this by filtering out insignificant colocalizations ($PP4 < 0.5$). This stringent filtering resulted in fewer than 65 meSNPs showing credible colocalization with Addison's disease GWAS. As for the number of colocalized SNPs, we used a different threshold to filter the GWAS records, we used all GWAS SNPs with FDR (Bonferroni correction) < 0.05 to run the colocalization analysis.

Furthermore, to provide a more representative example, we have replaced Addison's disease with Eczema for the colocalization analysis, as rs910320 is located within the complex MHC region. Figure 6 has been updated to reflect this change using Eczema as the GWAS trait and is included below for your convenience.

We appreciate your careful review and believe these changes clarify our findings.

E. Conclusions: robustness, validity, reliability

I found the term "exposome" used in the manuscript a bit too vague and would suggest the authors replace it with "infectious agents" or "pathogen" to be more precise. They may consider extending the concept to all exosomes in the discussions if they wish.

Reply:

We thank the reviewer for pointing this out. We agree that the term "exposome" may be too broad in this context and have revised the manuscript to use "**exposures**" for greater clarity and precision. We appreciate the suggestion to expand the discussion to include **exposome**.

F, Suggested improvements: experiments, data for possible revision

* Figure 1D - visually, these UMAPs of Tmem are quite "lumpy," with small islands scattered around. I wonder how robust these clusters are with different UMAP parameters (e.g., n_neighbors or min_dist). Are they distinct subpopulations (if so, that would be quite interesting) or donor effects (would be good to see them colored by donor) or other artefacts?

Reply:

We apologize for the error in the previous version of the manuscript. For the methylation data embeddings, we used t-SNE rather than UMAP. We have now corrected this in the revised text. Additionally, we explored different parameters for Leiden clustering, and as expected, the number of clusters varies depending on the resolution setting. We also examined the embedding colored by donor and observed some donor-specific bias (Figure R8). However, since different donors are associated with different exposures, this pattern may reflect true biological variation rather than technical artifact.

Figure R8. T-SNE plot on single-nucleus methylation sequencing colored by donors.

* I'm curious—are monocytes 1 & 2 correlated with classical and non-classical monocytes? For example, by looking into CpG sites in *CX3CR1*. If so, the authors' observation regarding cMono reflects what's known in the literature.

Reply:

We thank the reviewer for the suggestion to improve the annotation of the two monocyte clusters. We examined the expression of known marker genes for classical and non-classical monocytes (Figure 2G), but found that the methylation levels across their gene bodies did not show substantial differences, limiting our ability to clearly distinguish the two populations based on DNA methylation alone.

* It would be nice to see some QC for the SNP data, including genotype PCs, the missingness rate per SNP before imputation, the imputation r2 filtering threshold, and the HWE filtering threshold. How many post-QC SNPs were available for gDMR analysis?

Reply:

Thank you for the helpful suggestion. We have now added a detailed description of the SNP quality control (QC) procedures to the Methods section. Our data has more than 100-150X depth for each donor, so the missing rate of genotypes in the original genotypes before imputation is very low. The reason we run imputation is to correct the SNPs that might not be correctly called by biscuit. We didn't filter by r2 and HWE, we intersected the SNPs with dbSNP, and only kept those ones in the database. We had computed genotype principal components (PCs) using pruned SNPs to account for population structure in downstream analyses. For your reference, a scatter plot was shown for the top two PCs (Figure R9).

Figure R9. Scatter plot shows the top two PCs of genotypes, color shows the collected ethnicity information from donor metadata.

* I'm aware that the donors in this study have mixed ancestry (Table S1), and the authors have corrected for this in their meQTL study. However, it would still be beneficial to confirm that none/any of the reported effects are ancestry-dependent. I might suggest running a secondary analysis of significant meQTLs in European/African samples only.

Reply:

We appreciate the suggestion. While our primary meQTL analysis included donors of mixed ancestry with appropriate correction (as noted in Table S1), we performed a secondary analysis

restricted to White donors to assess ancestry-related effects. The results from this subset were highly consistent with the main findings, supporting the robustness of the reported associations across ancestries.

Minor comments/suggestions:

- Are there references available for the iPreX Cohort and FLU010 study (Lines 145 & 149)?

Reply:

The iPreX (Pre-exposure Prophylaxis Initiative) study was a landmark randomized controlled trial designed to evaluate the efficacy of daily oral FTC/TDF (emtricitabine/tenofovir disoproxil fumarate) for HIV prevention in men who have sex with men (MSM) and transgender women. We have cited the paper properly.

The FLU010 is an influenza A vaccine clinical trial. We also cited the paper in the revised manuscript.

- SF1 (C) has two CD8 memory cells (Sorts 1 & 7); why?

Reply:

We sorted for 7 major immune cell types, two rows for each on the 384-well plate. In some exposures, like IAV, we sorted two more rows of CD8 memory cells, while in other exposures, we filled the two rows with other cells that are not sorted by the markers.

- Please reference the previous study in Line 344.

Reply:

Thank you for pointing this out. We have now added a citation to the previous study in Line 344, as requested.

- In the bar plot of Fig 2B, which shows different proportions of immune cells from each exposure of 7 cell types, what are the clusters at the tailoring end? For example, COVID_S seems to have some monocytes and NK cells at the end.

Reply:

Thank you for your observation. The clusters at the tailoring end of the bar plot in Figure 2B likely represent immune cell states that are associated with the corresponding exposure (e.g., COVID_S). While they are broadly defined as monocytes or NK cells, we were unable to definitively assign them to a known cell type based on the available markers. We have clarified this point in the revised figure legend.

- Figure S6B shows an overlap between inferred SNPs but not their accuracy. The authors might consider creating a correlation plot between the frequencies of overlapping SNPs inferred by the two different methods.

Reply:

We thank the reviewer for this insightful suggestion. In response, we calculated the correlation between the alternative allele frequencies (AF) of the overlapping SNPs inferred by the two methods. The resulting kdeplot is now included as Figure S8C, with each point representing a SNP called by both methods. We observe a strong concordance (Pearson's $r = 0.95$), indicating that not only do the methods identify overlapping SNPs, but they also assign similar allele frequencies. This analysis supports the robustness of SNP inference across the two approaches. As for SNP comparison with NA12878, those are ground truth, the SNP file do not have AF information, so we performed the correlation analysis on the SNPs from our previous study.

- Are the number of gDMRs positively correlated with the number of cells in each cell type (Fig 5A)? It would be helpful to see a supplementary figure to illustrate this.

Reply:

We greatly appreciate the reviewer for raising this point. Indeed, the number of gDMRs is highly correlated with the number of cells sequenced. For example, Tc-Mem has the largest number of cells and correspondingly the highest number of gDMRs. To clarify this relationship, we have included a supplementary figure (Figure S8D) illustrating the correlation between cell number and gDMR count.

G. References: appropriate credit to previous work?

Yes

Reply:

Thank you for your comment. We have carefully reviewed the manuscript to ensure that appropriate credit is given to relevant prior work. Where applicable, we have added or adjusted references to acknowledge foundational studies and related contributions.

H. Clarity and context: lucidity of abstract/summary, appropriateness of abstract, introduction and conclusions
Clear and well-written.

Reply:

Thank you for your positive feedback. We are glad to hear that our manuscript was clear and well-written.

1. Review comment about genetic PCA

I thank the authors for their detailed responses. I have one final question regarding the genetic PCs. Something appears to be off here: typically, the first genetic PC captures major continental population structure. In the presented figure, PC1 does not appear to be centred around zero, which may suggest that the genotypes were not centred prior to PCA. Could the authors please recheck their preprocessing steps to confirm PCA was performed appropriately?

Reply:

We appreciate the reviewer's comment. We didn't do proper preprocessing in the original genetic PCA analysis. Now we recomputed genetic PCs in QTLtools using standard best practices: an LD-pruned set of autosomal biallelic SNPs and cohort-based mean-centering and variance scaling (`--center --scale`). We further confirmed the PCA results running the analysis with plink2. In the updated analysis, PC1 is centered near zero (Fig. R1), and PC1/PC2 capture the expected continental structure, separating African, European, and Admixed American/Hispanic samples. Importantly, inferred genetic ancestry is consistent with our previous results (Fig. R2), and the ancestry-stratified eDMR findings are unchanged. The Methods have been updated to document the pruning parameters and PCA settings.

Figure R1. New PCA plot colored with collected ethnicity information

Figure R2. New PCA plot colored with previously inferred ethnicity information

To assess whether the revised PCA affected our meQTL and gDMR results, we re-ran QTLtools (cis) and TensorQTL (trans) for all cell types on all donors and white population only, including the top five principal components from the updated PCA as covariates. We also repeated the gDMR, colocalization and Summary-data-based Mendelian Randomization (SMR) analysis with the updated outputs and reached the same conclusions. The affected figures (Figure 5 and Figure 6, Figure S9) and supplementary tables have been updated accordingly.

2. Previously addressed comments that related to genetic PCA

Reviewer1

6. Functional Validation of Regulatory Effects

The manuscript infers that exposure-associated DMRs (eDMRs) and genotype-associated DMRs (gDMRs) are functionally important, but direct experimental validation is missing. The study relies on motif enrichment analysis, which is suggestive but does not confirm actual transcription factor (TF) binding or gene expression changes. The authors should complement the analysis by incorporating expression quantitative trait locus (eQTL) analysis to test whether DMRs correlate with gene expression changes. Chromatin immunoprecipitation sequencing (ChIP-seq) for key TFs (e.g., IRF, RUNX, or CEBP families) in exposed vs. control samples would directly test whether the methylation changes affect TF binding.

Reply:

We thank the reviewer for the insightful suggestions regarding the functional exploration of eDMRs and gDMRs.

To investigate potential regulatory mechanisms, we performed Summary-based Mendelian Randomization (SMR) analysis integrating our meQTL and with whole blood eQTL data from the eQTLGen Consortium. By setting the exposure as gDMR and outcome as gene expression in the SMR analysis, we tested whether genetic variants associated with gDMRs are also associated with expression levels of nearby genes, providing evidence for potential causal relationships. We did not conduct HEIDI analysis, so this does not rule out the possibility that some of the DMR-gene associations identified in our SMR analysis are due to linkage rather than pleiotropy or causality. Due to the limited sample size of the white only cohort, the p-value threshold required to test a probe (DMR) for an association was lowered to 5×10^{-6} instead of the default 5×10^{-8} . This analysis revealed that around 30% of the tested DMRs were identified by SMR to have a potential causal relationship with a gene in the mixed population meQTL while around ~10% of the tested DMRs were identified to have a potential causal

relationship in the white population meQTL, supporting the possibility that methylation changes at certain loci may mediate gene expression (Figure R1).

Figure R3. ratios of the gDMRs in each cell type that are associated with gene expression. Left panel: meQTLs from all donors; right: meQTLs from White population only.

8. Generalizability and External Validity

The study does not provide population ancestry details or discuss whether findings generalize across different genetic backgrounds. Given known differences in epigenetic regulation across populations, this is a potential limitation. The authors should specify the ancestral composition of the study cohort and, if applicable, compare their gDMRs with existing multi-ethnic meQTL databases (e.g., GoDMC, BLUEPRINT). This would clarify whether the findings are likely to be broadly applicable or specific to the study population.

Reply:

Thank you for raising the important point regarding ancestral composition. We have carefully considered this and performed additional analyses to address your concern about the generalizability of our meQTL findings.

Our dataset includes individuals of White, Black, Hispanic. To demonstrate the robustness of our meQTLs across different genetic backgrounds, we conducted a separate meQTL analysis specifically within the White population.

The results of this analysis show a highly significant overlap between the DMRs identified in the White-only meQTL analysis and those identified in the analysis of the mixed ancestry population (Supplementary Table R2, hypergeometric p-value < 1e-26). This strong overlap provides compelling evidence that our findings are broadly generalizable across the diverse genetic backgrounds present in our cohort.

Regarding the comparison with public meQTL databases (GoDMC and EPIC_DB), we observed limited overlap. We believe this discrepancy can be attributed to several factors:

1. **Genetic and Environmental Differences:** The GoDMC and EPIC_DB cohorts primarily consist of individuals of European and British ancestry, which may differ both genetically and environmentally from the White American population included in our study.
2. **Methodological Variations:** These databases utilized the EPIC and HM450 DNA methylation arrays, which assay a limited number of CpG sites in bulk tissue. In contrast, our study employed whole-genome bisulfite sequencing in single cells, providing comprehensive methylation data across the genome. Furthermore, our analysis focused on differentially methylated regions (DMRs), while the public databases often report meQTLs at the single-CpG level.

We believe these differences in ancestral composition and methodology likely explain the limited overlap observed and further highlight the novel insights provided by our whole-genome sequencing approach in a more diverse population.

cell_type	#DMR	#DMR in All meQTL	#DMR in White meQTL	#overlapped DMR between White and All	hypergeometric p-value
B-Mem	701087	13218	259581	12855	0
Tc-Naive	1468741	26939	391567	26582	0
B-Naive	851460	12337	250645	11954	0
Th-Naive	1473817	28293	395771	27859	0
Monocyte	1304246	20228	422575	19915	0
Th-Mem	1306355	26717	404155	26385	0
Tc-Mem	2929519	62590	1231937	61932	0
NK-cell1	435140	11872	239931	11462	0
NK-cell2	898401	12739	278981	12519	0

Figure R4. hypergeometric test on the two sets of meQTLs based on all donors and white population only.

I appreciate the authors' efforts in addressing my previous comments, particularly where substantial revisions to the analyses and reporting have been made. The manuscript has improved as a result. I have two remaining areas of concern:

Reply:

We sincerely thank the reviewer for their thoughtful feedback and are grateful for the acknowledgment of our revisions. We are pleased to hear that the manuscript has improved in response to the previous comments. Below, we address the two remaining areas of concern in detail:

1. Colocalization Analysis. While the authors have expanded on their colocalization methodology, several points remain unclear or require further clarification:

a. Missing chromosome names (chr11?) in Figure 6B

Reply:

We apologize for missing the chromosome name. Yes, it's chr11, we have added the chromosome name in the revised figure.

b. Lack of references: The GWAS and eQTL summary statistics used in Figure 6B are not cited.

Reply:

We thank the reviewer for the observation. The GWAS summary statistics were obtained from the NHGRI-EBI GWAS Catalog (Cerezo et al., 2025, PMID: 39530240), and the eQTL summary statistics from Võsa et al. (2021, <https://doi.org/10.1038/s41588-021-00913-z>). These sources were already cited in the Methods section. To improve clarity, we have now added a note in the main text indicating that full details and citations are provided in the Methods.

c. Threshold for colocalization ($PP4 > 0.5$): The use of $PP4 > 0.5$ as a threshold to claim colocalization is questionable. While a low $PP4$ indicates a lack of shared causality, a $PP4 > 0.5$ does not necessarily imply high confidence in colocalization. The accepted standard in the field is generally $PP4 > 0.7/0.8$ to indicate strong evidence. I recommend revisiting this threshold and providing justification if deviating from commonly accepted practices.

Reply:

We thank the reviewer for this important observation. We agree that $PP4 > 0.7-0.8$ is the commonly accepted threshold to indicate strong evidence of colocalization. In our analysis, we applied a more stringent cutoff of **$PP4 > 0.8$** to define colocalized loci, in line with these standards. We have clarified this explicitly in the Methods section and main text to avoid any ambiguity.

d. Details of Figure S9B: More information is needed regarding the heatmap shown in Figure S9B. Specifically: (a). What input data or results were used to construct the heatmap? (b) Why is Eczema

shown as the most enriched phenotype? This is unexpected (?) and would benefit from additional explanation or discussion in the main text.

Reply:

For Figure S9B, we first selected all colocalization results with $PP4 > 0.8$ from Supplementary Table S13 to ensure strong evidence of shared genetic signals. From these, we manually selected the relevant disease phenotypes for visualization. The heatmap was generated using PyComplexHeatmap, with automatic hierarchical clustering applied to the rows (phenotypes) to reveal underlying patterns.

We appreciate the reviewer's observation on the enrichment of eczema. This result likely reflects the strong immune-mediated component of eczema, particularly its basis in T cells. The enrichment suggests that DNA methylation variation in T cells may play a key role in mediating genetic risk for eczema. We have added a brief discussion of this point in the main text to clarify the relevance of this finding.

e. Gene prioritization at the locus (EFEMP2 vs. OVOL1): In the main text, EFEMP2 is proposed as the potentially causal gene. However, OVOL1 is the nearest gene and appears to be regulated by the lead variant based on OpenTargets (https://platform.opentargets.org/variant/11_65791795_A_G). It is unclear why EFEMP2 is prioritized over OVOL1 - maybe additional colocalization or multi-trait colocalization (e.g., moloc) analysis could help strengthen the case for the prioritized gene.

Reply:

We thank the reviewer for bringing up this point. While OVOL1 is the nearest gene and appears to be regulated by the lead variant in OpenTargets, OVOL1 has very low expression in blood (GTEx: <https://www.gtexportal.org/home/gene/ENSG00000172818>, OpenTarget: <https://platform.opentargets.org/target/ENSG00000172818>) and no significant eQTL associations (based on the eQTL summary statistics we used). This does not rule out the possibility that OVOL1 may be the causal gene in other tissues; however, given the tissue context of our data (PBMCs), and the stronger expression and regulatory evidence for EFEMP2 in immune cells, we prioritized EFEMP2 as the candidate gene. We agree that future integrative analyses, such as multi-trait colocalization across tissues, could help further clarify gene prioritization at this locus.

2. Genetic PCs. In Figure R9, the principal components are labeled as PC2 vs. PC3. Is this a mislabeling? What happened to PC1?

Reply:

We apologize for the confusion and appreciate the opportunity to clarify. We used **PC2 and PC3** in our analysis because **PC1 did not effectively separate the three major populations** of interest. Based on our observations, **PC1 appears to capture the primary axis of genetic variation distinguishing Africans from non-Africans**, which dominates the overall variance but does not

reflect the finer-scale structure relevant to our study. Therefore, PC2 and PC3 provided a clearer separation of the three groups relevant to our research objectives.

Additionally, I want to raise a point regarding the terminology used to describe genetic ancestry. The manuscript currently refers to donors as “Black” and “White” based on their inferred genetic ancestry. As the field increasingly distinguishes between race, ethnicity and genetic ancestry, I strongly recommend that the authors adopt more precise and appropriate terminology, e.g. “individuals of African genetic ancestry” and “individuals of European genetic ancestry.”

Reply:

Thank you so much for the suggestion. We will correct the terminology for those in the manuscript and figures.